# FLIP(C1orf112)-FIGNL1 complex regulates RAD51 chromatin association to promote viability after replication stress

Jessica D. Tischler[1,3], Hiroshi Tsuchida[1,3], Rosevalentine Bosire[1], Tommy T. Oda[1,2], Ana Park[1,2] & Richard O. Adeyemi [1,1] ✉

Homologous recombination (HR) plays critical roles in repairing lesions that arise during DNA replication and is thus essential for viability. RAD51 plays important roles during replication and HR, however, how RAD51 is regulated downstream of nucleofilament formation and how the varied RAD51 functions are regulated is not clear. We have investigated the protein c1orf112/FLIP that previously scored in genome-wide screens for mediators of DNA inter-strand crosslink (ICL) repair. Upon ICL agent exposure, FLIP loss leads to marked cell death, elevated chromosomal instability, increased micronuclei formation, altered cell cycle progression and increased DNA damage signaling. FLIP is recruited to damage foci and forms a complex with FIGNL1. Both proteins have epistatic roles in ICL repair, forming a stable complex. Mechanistically, FLIP loss leads to increased RAD51 amounts and foci on chromatin both with or without exogenous DNA damage, defective replication fork progression and reduced HR competency. We posit that FLIP is essential for limiting RAD51 levels on chromatin in the absence of damage and for RAD51 dissociation from nucleofilaments to properly complete HR. Failure to do so leads to replication slowing and inability to complete repair.

Cells are constantly exposed to agents that can damage DNA[1,2]. These problems can be particularly inimical during cellular replication during which failure to adequately repair the lesions formed can lead to severe cellular consequences including mutations, loss of genetic information, chromosomal fusion events, and so on, all of which can lead to various diseases including cancer[3–5]. Among the various types of lesions that can arise during replication, helix distorting crosslinks, especially inter-strand crosslinks (ICLs) in which two opposite strands are covalently ligated, can pose an absolute impediment to replication and/or transcription complex advancement and are particularly cytotoxic[6]. ICLs can arise as byproducts of endogenous metabolic events, for example, during aldehyde and nitrous acid metabolism[7,8]. Several platinum-based ICL-inducing agents such as cisplatin, oxaliplatin and mitomycin C (MMC) enjoy widespread use in chemotherapy owing to the exquisite sensitivity of rapidly dividing cells to these drugs[9,10].

Crosslink repair is typically coupled to replication in eukaryotes, and it involves the concerted action of various repair pathways[6]. Among the major players are the nucleotide excision repair (NER) pathway, translesion synthesis (TLS) and HR pathways. The Fanconi anemia (FA) pathway comprises several HR genes as well as multiple genes that make up the FA core complex which, upon sensing the ICL, activates a critical complex of FANCI and FANCD2 proteins (the I-D complex) via mono-ubiquitination by the ubiquitin ligase FANCL[11,12]. This step is necessary for the incision events that unhook the ICL prior to NER, TLS and HR. Germline mutations affecting genes in the FA pathway cause Fanconi anemia, a rare genetic disorder characterized by bone marrow suppression, hematopoietic and growth defects and increased cancer predisposition[13–15].

A critical step in HR is the efficient nucleation of single-stranded DNA (ssDNA) by the recombinase RAD51 to mediate homology search,

[1]Basic Sciences Division, Fred Hutchinson Cancer Center, Seattle, WA 98109, USA. [2]University of Washington, Seattle 98195, USA. [3]These authors contributed equally: Jessica D. Tischler, Hiroshi Tsuchida. ✉e-mail: radeyemi@fredhutch.org

strand invasion and pairing, prior to copying of homologous strands for repair[16,17]. Because of its highly recombinogenic potential, this process is tightly regulated, with several factors called mediators (such as BRCA2 and the RAD51 paralogs) acting to displace the ssDNA-bound replication protein A (RPA) and promote RAD51 nucleofilament formation while others like the BLM, PARI and RECQL5 oppose and/or fine-tune the process[18–22]. Downstream of filament formation, nucleofilament disassembly and subsequent DNA synthesis is also a critical but not well understood process, and only recently have important new players such as ZGRF1 and HROB (MCM8IP) been identified to promote RAD51 filament disassembly and allow postsynaptic synthesis, providing more insight into these latter steps of HR[23–25]. All of these factors are essential for viability in response to ICL treatment.

In addition to classical HR functions of RAD51, recent work has highlighted break-repair independent functions for RAD51 during stalled fork metabolism[26–29]. These range from promoting fork reversal, a process in which stalled forks are processed into four-way junctions to stabilize the forks, to directly protecting forks from degradation by exonucleases[26–30]. Indeed, RAD51 itself is a Fanconi gene, FANCR, and certain mutants of RAD51 that are competent for HR are still quite sensitive to ICLs[30], demonstrating that RAD51 plays multiple roles during ICL repair. RAD51 can bind to both ssDNA and dsDNA in vitro, and although binding to dsDNA is detrimental to its HR functions, such binding has recently been shown to be critical for maintaining fork integrity[31]. RAD51 has been shown to associate with DNA during normal replication in the absence of damage[27] and, while a few novel regulators of RAD51's HR function such as FIGNL1 has been identified[32], there remains a need to characterize the various factors that regulate the varied repair-independent functions of RAD51.

We and others previously identified c1orf112 from whole genome CRISPR screens for ICL sensitizers[33,34]. Here we have investigated the role of c1orf112, a previously poorly characterized factor that we now show is important for ICL repair and RAD51 regulation. We describe c1orf112/FLIP as a novel DNA repair factor that regulates RAD51 chromatin association in the absence and persistence on nucleofilaments upon induction of exogenous DNA damage. FLIP interacts with FIGNL1 through its N-terminal region and together both proteins form a stable complex. Upon replication stress, loss of FLIP leads to increased genomic instability, characterized by elevated damage signaling, chromosomal aberrations, micronuclei formation, and defective HR.

## Results

### C1orf112/FLIP is a novel ICL repair gene

We previously reported a genome-wide CRISPR/Cas9 knockout screen aimed at identifying novel genes required for ICL repair specifically and the replication stress response in general[33]. This screen generated a high confidence dataset of genes not previously linked to the replication stress response. The Durocher group also reported similar cisplatin screens performed in a different cell line (RPE1) as part of a wider analyses of DNA repair dependencies[34]. In order to mine high-confidence hits for follow up analysis, we selected the top-scoring 500 genes from two cisplatin screens done in RPE1 cells and two done in U2OS cells. To discover genes that function in a cell-type independent manner we looked for hits that scored in at least 3 screens. Our analysis revealed at least 70 genes, almost all of which have been linked to DNA repair (Fig. 1a, Supplementary Data 1). We decided to focus on c1orf112 (FLIP), one of the uncharacterized genes on the list. FLIP was one of the top 150 hits in both RPE1 cisplatin screens analyzed and was also in the top 3% of hits in PD5 of our U2OS cisplatin screens (Fig. 1a). The major isoform of FLIP encodes an 853 amino acid (aa) protein that is well conserved across vertebrates[35]. Mammalian FLIP is largely uncharacterized, however a few publications have linked high expression of FLIP to poor prognosis in various cancer types[36,37].

To validate a role for FLIP following DNA damage, we depleted FLIP in U2OS cells using two independent validated siRNAs and performed multicolor competition assays (MCAs) (Fig. 1b, c). This assay allows us to examine roles for FLIP following various genotoxic treatments. ATM (important for double strand break repair) and FANCD2 (an HR factor that is essential following various replication stress events) depletion served as controls. Whereas we failed to observe significant sensitivity following ionizing radiation (IR) treatment (which causes double strand breaks and base damage) using this assay, FLIP depletion significantly increased sensitivity to cisplatin treatment (Fig. 1b).

To further characterize FLIP's role, using CRISPR-Cas9 gene editing, we generated two knockout clones in U2OS cells (Supplementary Fig. 1A) and, using clonogenic survival assays (CSAs), examined the knockouts for sensitivity to multiple ICL agents, hydroxyurea (HU, to examine replication stress sensitivity) and IR. Both knockout clones showed similar increased sensitivity to mitomycin C (MMC, another ICL-inducing agent) (Fig. 1d). We also confirmed cisplatin sensitivity using CSAs (Supplementary Fig. 1B). In addition to ICL agents, we observed significantly increased sensitivity to HU (Fig. 1e) and slight but consistent sensitivity to IR (Fig. 1f). These data, and the mild IR sensitivity suggest a general requirement of FLIP following replication stress or in break repair. To further define roles for FLIP in additional cell lines, we depleted FLIP using siRNAs in HeLa cells and RPE1 cells and examined cellular viability following cisplatin treatment using CSAs. Knockdown of FLIP led to significantly reduced viability in both of these cell types following treatment with the ICL-inducing agent cisplatin (Supplementary Fig. 1C, D). Taken together, these results from multiple cell lines demonstrate the essential function of FLIP in promoting cell viability after treatment with ICL and replication stress inducing agents.

### FLIP prevents genomic instability following replication stress

To further characterize repair roles for FLIP upon ICL agent treatment, we assayed for cell cycle alterations. FLIP loss did not cause significant changes in cell cycle distribution in the absence of damage (Supplementary Fig. 1E, F), although there was slight accumulation of cells in the G2/M population. Upon cisplatin treatment however, loss of FLIP led to marked alterations in cell cycle progression. In these experiments, cells were treated with cisplatin for a 24 h period prior to washout to allow the cells to recover for up to 2 days after drug treatment (Fig. 1g). Whereas control cells showed mild accumulation of cells initially in S (0 h) and later in G2 (24 h after release) with almost complete recovery by day 2 after drug treatment, loss of FLIP led to marked increases in the S and G2 accumulated populations that progressed to significant apoptosis (sub G1 population) 2 days after drug treatment (Fig. 1h, i).

Next, we performed metaphase spreads to examine cells for genomic instability upon FLIP loss (Fig. 2a, b). FLIP knockdown led to significant increases in spontaneous chromosomal abnormalities (quantified in Fig. 2b, left panel). To test the effect of ICL treatment, we employed low MMC doses in which very little chromosomal aberrations arose in control cells. Upon MMC treatment, FLIP knockdown significantly increased the amounts of chromosomal abnormalities−breaks and gaps (Fig. 2a, b, right panel). One characteristic chromosomal abnormality seen upon loss of Fanconi genes following ICL treatment is radial formation[7,38]. As expected, knockdown of the Fanconi gene FANCD2 led to significant increases in radial formation. However, at the dose of MMC used in our experiments (2 ng/ml), we did not observe such increases in radial formation upon loss of FLIP (Supplementary Fig. 2A), suggesting that although FLIP is important for ICL repair, it may not do so as a member of the Fanconi pathway.

Micronuclei formation is another hallmark of replication stress[39]. They form as a result of lagging and/or acentric chromosomes that are not incorporated into daughter cell nuclei during cell division[39,40]. We treated cells with cisplatin and allowed cells to recover for 24 h before

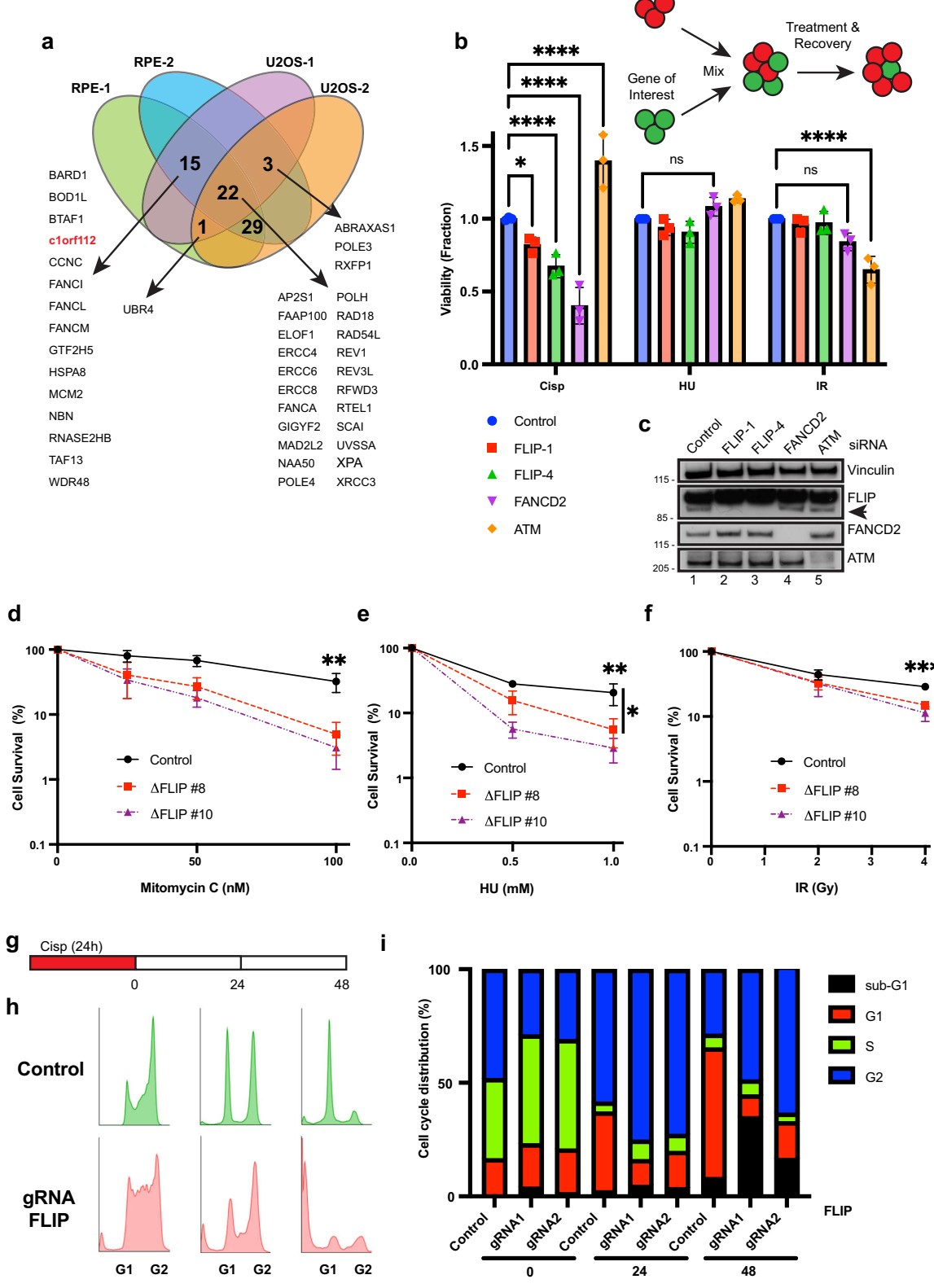

assaying for the number of micronuclei formed per cell. FLIP knockdown led to increased frequency of nuclei with higher numbers of micronuclei associated with them compared to controls (Fig. 2c). There was no such increase in the absence of drug treatment. Taken together, our results show that FLIP is essential for maintaining genomic stability especially after treatment with ICL agents.

**FLIP promotes ICL repair independently of the Fanconi pathway**

The striking requirement for FLIP following treatment with agents that induce DNA crosslinks prompted us to investigate whether FLIP modulates the Fanconi pathway in some way, since several Fanconi genes play prominent roles in promoting cell viability following ICL induction. During repair of DNA crosslinks, FANCM senses DNA lesions

**Fig. 1 | FLIP is important for ICL repair. a** Schematic showing overlap of high-priority hits from published cisplatin screens. Genes that scored in at least three screens are indicated. FLIP (c1orf112) is indicated in red. All 5 overlaps are listed except for the 29 genes due to space limitations. **b** Top: schematic of multicolor competition assay (MCA). Middle and bottom: U2OS-GFP cells were reverse transfected and processed as described in the methods section. MCA showing survival of U2OS cells after treatment with the indicated drugs and siRNAs. Data are normalized to untreated cells, mean ± SD shown, $n = 3$ independent experiments, statistics represent two-way ANOVA followed by Dunnett's multiple comparison test, ns − not significant; *$P = 0.0327$; ****$P \leq 0.0001$. Doses and duration of drugs were 1 µM for 24 h using cisplatin and 2 mM for 24 h using hydroxyurea (HU).

Ionizing radiation (IR) was performed using 9 Gy. **c** Western blots (WB) confirming knockdown of indicated proteins in the MCA assay. FLIP protein is indicated with an arrow below the non-specific band. CSAs showing survival of U2OS cells expressing control gRNA or two FLIP gRNA knockout clones upon 24 h treatment with indicated doses of MMC in (**d**), HU in (**e**) and IR in (**f**). For **d**−**f**, mean ± SD shown, $n = 3$ independent experiments, *$P \leq 0.0135$; **$P \leq 0.01$; ***$P \leq 0.001$, Ordinary one-way ANOVA followed by Dunnett's of the indicated dose. **g** Schematic of cell cycle assay. Cisplatin was added at 2 µM final concentration. Cells were washed and allowed to recover for the indicated times. **h, i** Cell cycle distribution in control or FLIP gRNA expressing cells at the indicated times following cisplatin treatment. $N = 3$, representative experiment shown. Source data are provided as a Source Data file.

and recruits the Fanconi complex, consisting of several genes that activate FANCI-FANCD2 via phosphorylation and mono-ubiquitination[11,41]. In order to examine whether FLIP functions as part of the Fanconi pathway, we assayed WT and ΔFLIP cells for FANCI foci. Following MMC treatment, there were increased amounts of γH2AX as well as FANCI foci (Supplementary Fig. 2B, C). However, we observed no significant reduction in the number of FANCI foci upon FLIP loss (Supplementary Fig. 2B, C). To further determine whether FLIP modulates Fanconi pathway activation, we performed immunoblotting to directly assay for mono-ubiquitination of FANCD2, seen as a slight reduction in protein mobility. Cisplatin treatment led to increased mono-ubiquitination of FANCD2, with no apparent reduction in this process upon FLIP depletion (Supplementary Fig. 2D, compare lanes 5 and 6 to lane 4).

To further characterize FLIP's potential interaction with the Fanconi pathway, we depleted FANCA in WT or ΔFLIP cells. FANCA is core member of the complex that mono-ubiquitinates and activates FANCI-D2, thus FANCA depleted cells are very sensitive to treatment with ICL agents. Near complete FANCA depletion was confirmed by western blotting (Fig. 2e). ΔFLIP cells showed similar sensitivity to ICLs as compared FANCA knockdown cells, with ΔFLIP cells showing slightly higher sensitivities as the dose of drug increased (Fig. 2d). Importantly, co-depletion of FANCA and FLIP led to increased reduction in viability compared to depletion of either protein alone (Fig. 2d, e). Taken together, while FLIP plays vital roles in promoting cellular viability after ICL treatment, it appears to do so in a manner that is distinct from but additive to the roles played by the FA pathway.

### FLIP is recruited to DNA damage sites and promotes repair following DNA damage

To visualize FLIP localization following DNA damage we performed immunofluorescence (IF) assays using antibodies against epitope−tagged FLIP following cisplatin treatment. FLIP formed multiple foci upon cisplatin treatment (Fig. 3a, b). Similar results were seen in both U2OS and HeLa cells (Supplementary Fig. 3A). There were some vehicle-treated cells that showed FLIP foci formation (Fig. 3b). FLIP foci only partially colocalized with γH2AX (Fig. 3a, b), and in many cases was situated right adjacent to γH2AX foci. Knockdown of FANCD2 almost completely abolished foci formation by FLIP after cisplatin treatment (Fig. 3c, d), despite little to no colocalization with FANCD2 (Supplementary Fig. 3B), suggesting that FLIP acts downstream of FANC-ID foci formation but upstream of CtIP, whose knockdown did not prevent FLIP foci formation (Fig. 3c, d).

Next, we examined for any differences in DNA damage signaling upon FLIP loss. During replication stress, RPA-coated ssDNA at stalled replication fork junctions trigger activation of ATR[42–44]. This initiates a signaling cascade in which several proteins ranging from repair genes to checkpoint modulator are activated. FLIP knockdown led to altered DNA damage signaling characterized by increased phosphorylation of the checkpoint protein Chk1 as well as increased RPA phosphorylation following cisplatin treatment (Fig. 3e). FLIP knockdown also consistently led to increased γH2AX amounts following cisplatin treatment (Fig. 3f). This was seen both following gRNA and siRNA depletion

of FLIP (Fig. 3f), with multiple gRNAs and siRNAs showing effects correlating to the level of FLIP knockdown.

The increased chromosomal aberrations observed in the absence of FLIP (Fig. 2b) suggests that FLIP is important for regulating break repair or break formation (or both). To further characterize FLIP's role in preserving genomic integrity, we performed Comet assays. RADX, whose loss causes break formation in the absence of exogenous damage, served as a control. Similar to RADX, FLIP loss led to increased break formation and/or persistence (Fig. 3g). Taken together, these data reveal that FLIP is important for repair after endogenous or exogenous sources of DNA damage.

### FLIP interacts with FIGNL1

BioGrid interactomes obtained from published mass spectrometry datasets suggest interactions between FLIP and FIGNL1[45]. FIGNL1 was originally identified in a proteomics dataset for interactors of RAD51[32]. Examining the reported IP mass spec dataset for interactors of FIGNL1 showed multiple peptides to c1orf112/FLIP. Indeed, the *Arabidopsis* ortholog of FLIP shows interaction between the two proteins[46]. To determine whether both mammalian proteins interact, we expressed GFP-tagged FLIP and FLAG-tagged FIGNL1 in 293 cells. Using co-immunoprecipitation (coIP) experiments, FIGNL1 was able to pull down tagged FLIP, validating their interaction (Fig. 4a). Likewise, expression of tagged-FLIP pulled down the endogenous FIGNL1 protein (Fig. 4b). Notably, the interaction was constitutive, and did not require treatment with DNA damaging agents. To determine whether DNA damage signaling enhanced the interaction between the two proteins, we performed the coIPs in the presence or absence of cisplatin. While we consistently observed slight reduction in the amounts of FLIP pulled down by FIGNL1 following cisplatin treatment (Fig. 4c), we did not see such reduction when the reverse experiment was performed and FLIP was used to pull down FIGNL1 (Supplementary Fig. 4A). The reason for this difference is not yet clear.

Next, to map the domain that mediates binding between FLIP and FIGNL1, we generated a series of N and C-terminal truncation mutants of FLIP. A few of these mutants were unstable and could not be expressed (summarized in Fig. 4f). However, our experiments revealed that a mutant lacking the N-terminal 227 aa was severely compromised in its ability to bind to FIGNL1 (Fig. 4d), suggesting that binding between FLIP and FIGNL1 requires this N-terminal region. Further deletions beyond the first 350 aa of the protein completely abolished the interaction (Fig. 4d). Although the N-terminal 227 aa was required, a truncation mutant expressing only the N-terminal portion of the protein was unstable, precluding us from determining whether the N-terminal region was sufficient to bind to FIGNL1. (These C-terminal truncations were tagged with an NLS, since the predicted NLS of FLIP is located at the C-terminus of the protein ruling out mislocalization as the reason for the instability of the protein).

To map the interaction region of FLIP on FIGNL1, we started by generating two previously reported mutants that divide the FIGNL1 protein into an N-terminal 120aa region and a large truncation lacking the N-terminus (Fig. 4f). This latter mutant was previously shown to be

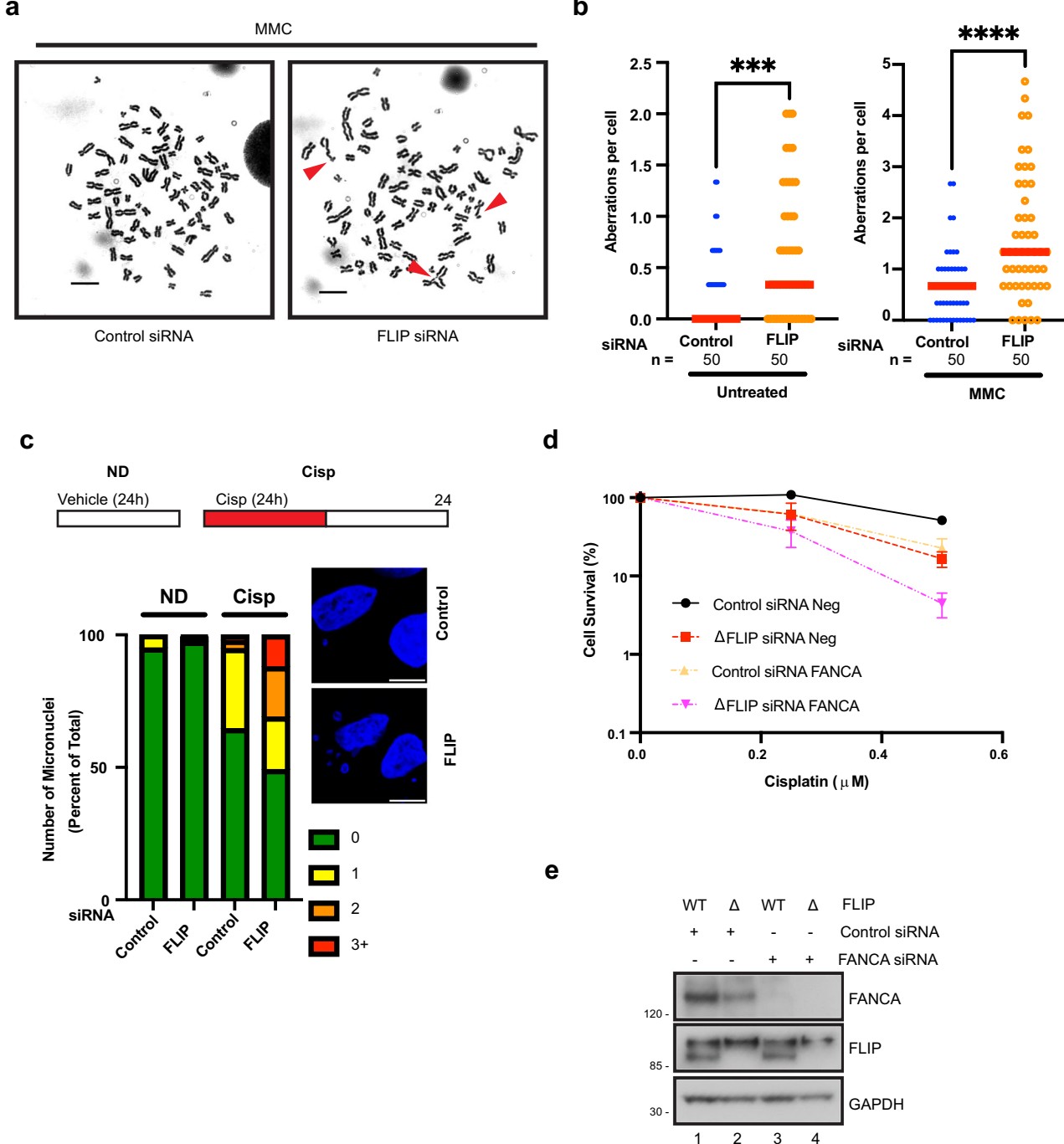

**Fig. 2 | FLIP prevents genomic instability after replication stress and functions downstream of Fanconi pathway activation. a** Representative images showing increased genomic instability in MMC-treated, HeLa cells following FLIP siRNA compared to control siRNA. Aberrations are shown in red arrows. **b** Quantification of total aberrations from untreated cells (left panel) and the experiment shown in (**a**), right panel, treated with 2 ng/ml MMC, n = 50, 3 independent experiments, red line represents median. ***P = 0.0009; ****P ≤ 0.0001, two-tailed Mann–Whitney tests. **c** U2OS cells treated with control or FLIP siRNA were exposed or not to 1 μM cisplatin for 24 h. Number of micronuclei per cell was quantified by DAPI staining 24 h after release from cisplatin treatment. ND – no drug, n = 3, representative experiment shown. **d** U2OS cells expressing control gRNA (WT) or ΔFLIP cells were treated with control or FANCA siRNA for 48 h. Cells were then exposed to the indicated doses of cisplatin and processed for CSAs as above. Mean ± SEM shown, n = 3 independent experiments. **e** Western blots showing knockdown of the indicated proteins from the experiment in (**d**). Scale bars = 10 μm. Source data are provided as a Source Data file.

defective in FIGNL1's HR functions upon DNA damage. In addition to 3X-FLAG tags, the N-terminal region was tagged with an SV40-dervied NLS, since this mutant was previously shown to localize to the cytoplasm when expressed unlike the full-length protein and the C120 mutant that both localize to the nucleus. Expressing the three proteins in 293 cells, coIP experiments revealed a striking loss of interaction upon deletion of the N-terminal region (Fig. 4e). Importantly, expression of only this region was sufficient to pull down FLIP (Fig. 4f) although at reduced levels, as this mutant was expressed at much lower levels than the full-length protein. Taken together, we conclude that binding between FIGNL1 and FLIP required their respective N-terminal regions.

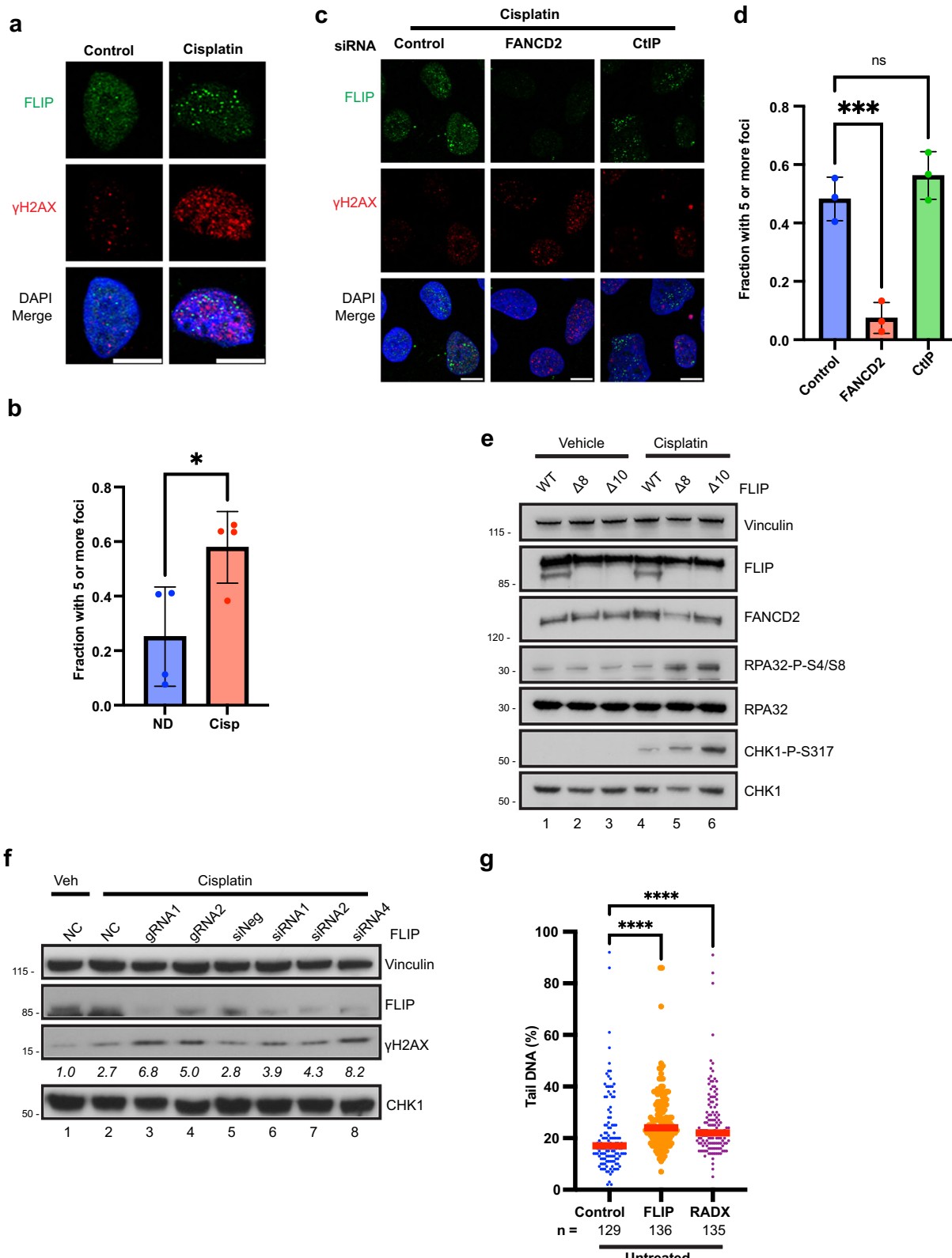

**FLIP / FIGNL1 function in the same pathway to promote viability after ICL treatment**

Both FLIP and FIGNL1 scored in genome-wide cisplatin screens, and DepMap analysis for dependents of FLIP showed a high correlation between loss of FLIP and loss of FIGNL1[47], suggesting that both proteins not only interact but likely functioned in the same pathway. Indeed, the N-terminal deficient mutant of FIGNL1 was previously shown to be unable to rescue HR defects upon FIGNL1 depletion and was not recruited to DNA damage sites, suggesting that FIGNL1 loss may be epistatic with FLIP loss[32]. To test this, we depleted FIGNL1 using siRNAs in WT vs FLIP knockout cells and examined sensitivity to cisplatin by CSAs. Loss of FIGNL1 led to increased sensitivity to cisplatin that was almost as high as FLIP knockouts (Fig. 5a). Importantly, unlike what was seen with FANCA depletion (Fig. 2g, h), depletion of FIGNL1 in

**Fig. 3 | FLIP is recruited to damage foci and limits DNA damage signaling after ICL treatment. a** IF images showing recruitment of GFP-FLIP to foci that partially colocalize with γH2AX after treatment with 0.5 µM cisplatin for 16 h. Cells were pre-extracted to remove soluble protein. **b** Quantification of the experiment in (**a**). $N = 4$ independent experiments, mean ± SD shown. *$P = 0.0268$, unpaired $t$ test, two-tailed. **c, d** U2OS cells were transfected with the indicated siRNAs for 48 h prior to treatment with 0.5 µM cisplatin for 16 h. IF images show reduced recruitment of GFP-FLIP to foci following loss of FANCD2. Quantified in (**d**). $N = 3$ independent experiments, mean ± SD shown, ns – not significant ($P = 0.4041$), ***$P = 0.001$, One-way ANOVA followed by Tukey's multiple comparison's test. **e** Control gRNA expressing or two independent ΔFLIP U2OS clones were exposed to vehicle or 2 µM cisplatin for 18 h prior to blotting for the indicated proteins, $n = 3$ independent experiments. **f** U2OS cells expressing control or two independent gRNAs to FLIP (lane 1–4), or U2OS cells 48 h after reverse transfection with control siRNAs or three independent siRNAs to FLIP (lane 5–8) were treated with vehicle or 1.5 µM cisplatin for 16 h prior to western blotting for the indicated proteins. Blots show increased γH2AX levels following knockdown of FLIP using siRNAs and gRNAs. γH2AX numbers are normalized to vinculin loading control, $n = 2$ independent experiments. **g** U2OS cells were transfected with the indicated siRNAs and cells were processed for neutral comet assays as described in methods. Comets were quantified in image J and the percentage of the comet tail signal to the whole comet signal (Tail DNA %) was calculated. Red bar represents median, three independent experiments. ****$P \leq 0.0001$, Kruskal-Wallis test followed by Dunn's. Scale bars = 10 µm. Source data are provided as a Source Data file.

ΔFLIP cells was no different than control siRNA treatment, suggesting that both proteins function in the same pathway in mediating survival after replication stress (Fig. 5a).

To further determine whether both proteins functioned together, we reasoned that FLIP mutants that failed to interact with FIGNL1 might be defective in rescuing FLIP knockout cells compared to the full-length FLIP. To test this, we complemented ΔFLIP cells with vector, WT-FLIP as well as a FLIP mutant lacking the N-terminal 227aa, a mutant that we previously showed failed to interact with FIGNL1 (Fig. 4d). The FIGNL1 interaction-defective mutant of FLIP was unable to mediate resistance to cisplatin compared to the WT (Fig. 5b). While this data suggests that FLIP and FIGNL1 function in the same pathway to mediate cellular viability after exposure to replication stress agents like cisplatin, we cannot rule out that the N228 truncation of FLIP has additional important functions outside of FIGNL1-binding.

## FLIP and FIGNL1 form a stable complex

While performing the rescue experiments in Fig. 5b, we observed that FLIP null cells appeared to have undetectable FIGNL1 expression (Fig. 5c, compare lanes 1 and 2) that was only slightly rescued by re-expression of GFP-tagged full-length FLIP. This data, coupled with the fact that co-depletion of FIGNL1 in FLIP KO cells behaved similarly to control siRNA knockdown prompted us to examine the expression levels of both proteins when either protein was depleted. We repeatedly observed a striking loss of expression of the FIGNL1 protein in the FLIP null cells (Fig. 5d, compare lane 1 and 2). This was not an artifact peculiar to this knockout (KO) clone as two different KO clones showed similar losses in FIGNL1 expression (Fig. 5d, compare lanes 2 and 3 to lane 1). This suggested that FLIP stabilizes the FIGNL1 protein. Since gRNA cloning takes several weeks and could lead to compensatory genomic alterations in other genes, and to determine whether short term depletion of FLIP affected FIGNL1 levels, we depleted FLIP with two different siRNAs. FIGNL1 depletion served as a control. Acute depletion of FLIP using two different siRNAs led to reduced levels of FIGNL1 (Fig. 5e). To our surprise, siRNA depletion of FIGNL1 also led to loss of FLIP (Fig. 5e, lane 4), demonstrating that not only does FLIP stabilize FIGNL1, but both proteins appear to form a stable complex. However, the loss of FIGNL1 expression in the FLIP null cells was not absolute (Supplementary Fig. 5A, long exposure) and could not be restored by proteasomal inhibition.

## FLIP interacts with and regulates RAD51 association with and persistence on chromatin

FIGNL1 interacts with RAD51 through an FXXA domain that is present in several RAD51 binding proteins[32,48]. Since FLIP interacts with FIGNL1, and both proteins form a complex, we tested whether FLIP could also bind to RAD51. CoIP experiments using HA-tagged RAD51 and FLAG-tagged FLIP confirmed that FLIP interacts with RAD51 (Fig. 6a). The amount of RAD51 pulled down with FLIP was lower than with FIGNL1 coIP, however, FLIP was expressed at much lower levels (Fig. 6a, PreIP, compare lane 2 and 3). The interaction was resistant to benzonase treatment suggesting that it was not mediated by DNA. FLIP was also pulled down using RAD51 (Fig. 6b). FLIP appeared to be able to bind to RAD51 independently of FIGNL1, as siRNA depletion of FIGNL1 had no effect on the ability of FLIP to bind to RAD51 (Fig. 6c).

During repair, RAD51 loading onto ssDNA leads to detectable foci formation. To determine whether FLIP modulates this process of HR, we sought to determine whether FLIP regulated RAD51 foci formation. 24 h after cisplatin treatment (0 h after release), FLIP depletion did not lead to reduction in RAD51 foci formation (Fig. 6d, e, f). Instead, upon FLIP loss, we observed slight increases in the percentage of cells with greater than 10 RAD51 foci as well as increases in RAD51 nuclear staining (Fig. 6e, f). This was seen both in ΔFLIP cells (Fig. 6e, f) as well as upon siRNA depletion of FLIP (Supplementary Fig. 6A, B). Due to spontaneous DNA damage formation in cells undergoing replication stress, a few foci of RAD51 are often observed in the absence of exogenous DNA damage. Strikingly, unlike the mild increase seen upon drug treatment, we observed significant increases in the number of RAD51 foci upon FLIP loss in untreated cells (Fig. 6e, f).

To further characterize FLIP effects on RAD51 foci formation, cells treated with cisplatin for 24 h were allowed to recover for one or more days after drug washout. Interestingly, loss of FLIP led to persistence of RAD51 foci following cisplatin treatment (Fig. 6c, e). This was also seen after siRNA treatment to deplete FLIP (Supplementary Fig. 6A, B). RAD51 persisted at higher amounts in ΔFLIP cells even 3 days after release from treatment (Supplementary Fig. 6C, D), with the latter time points (48 h and 72 h) showing higher fold change increases in RAD51 foci staining in ΔFLIP cells compared to controls (average of 25% in controls and 60% in ΔFLIP at 48 h, 8.7% and 36% at 72 h), consistent with persistent RAD51 foci. These results suggest that FLIP may regulate dissociation of RAD51 from damage foci after DNA damage, providing a possible mechanism for DNA repair defects seen upon FLIP loss after ICL agent treatment.

As an alternative approach to examine the effects of FLIP on RAD51, we performed chromatin fractionation experiments in WT and ΔFLIP cells. We consistently observed increases in the amounts of RAD51 in chromatin fractions whenever FLIP was absent (Fig. 6h). This was not due to increases in RAD51 expression, as FLIP loss did not lead to downregulation of RAD51 levels in whole cell lysates (Supplementary Fig. 6E). Such increase in chromatin associated RAD51 after DNA damage was also observed after siRNA depletion of either FLIP or FIGNL1 (Supplementary Fig. 6F) and could be rescued by expressing tagged FLIP (Fig. 6i, compare lanes 2&3 or lanes 5&6).

## FLIP / FIGNL1 promote RAD51 dissociation from chromatin

FLIP's binding partner, FIGNL1, can dissociate RAD51 from nucleofilaments in vitro[49]. To determine whether FLIP actively promotes RAD51 removal from foci, and to rule out whether the increased RAD51 foci seen in the absence of FLIP was not merely a consequence of increased break formation (Fig. 2, 3), we took advantage of cells modestly over-expressing GFP-tagged FLIP (Fig. 7a). We reasoned that FLIP overexpression should promote RAD51 dissociation from foci regardless of γH2AX status. Indeed, whereas both FLIP knockout and FLIP-overexpressing cells (FLIP-OE) had

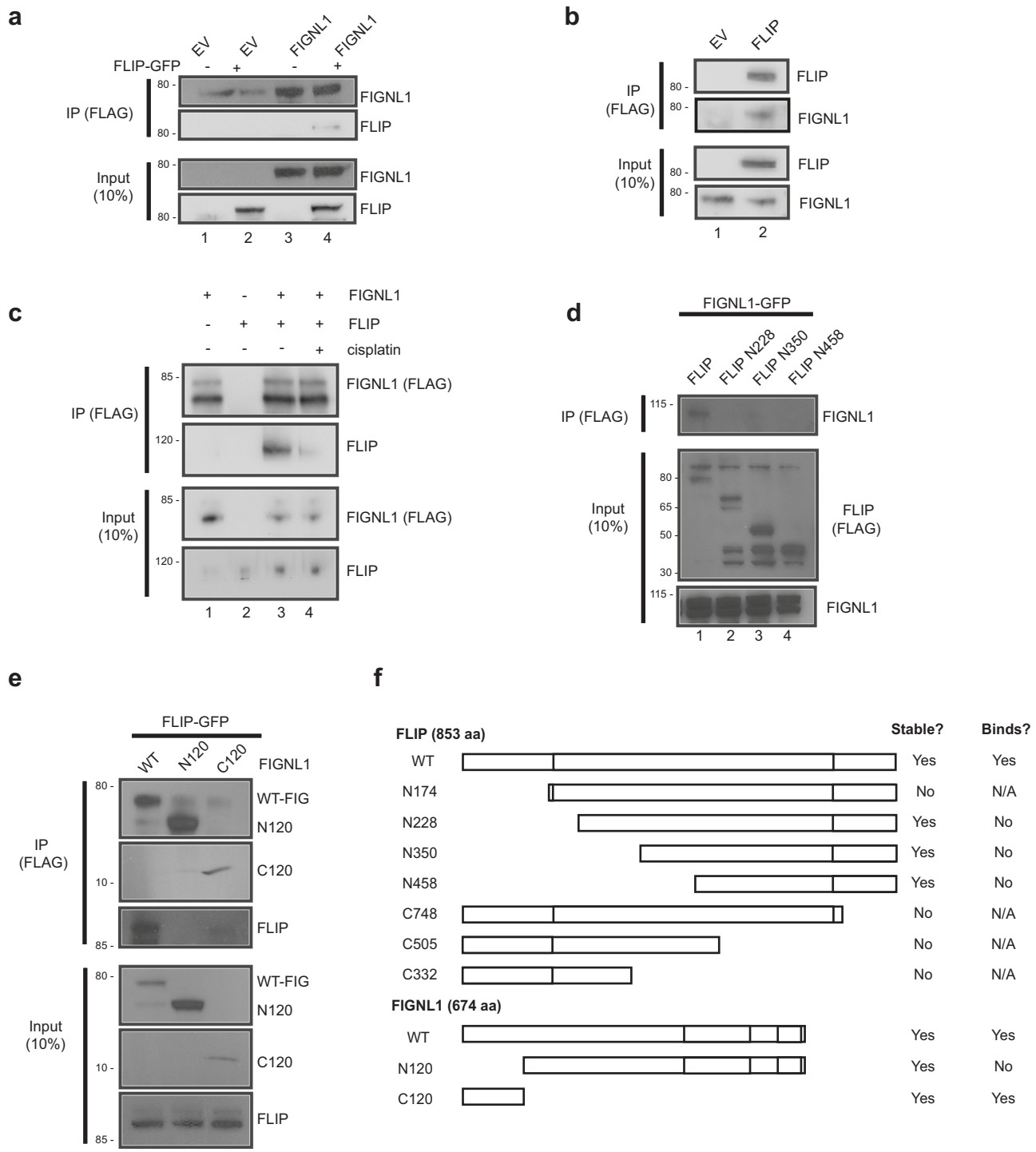

**Fig. 4 | FLIP forms a complex with FIGNL1. a** 293 T cells were transfected with empty vector (EV), FLAG-FIGNL1 or GFP-FLIP as indicated for 48 h. Blots show association of FIGNL1 and FLIP, *n* = 3 independent experiments. **b** 293 T cells were transfected with EV or FLAG-FLIP for 48 h. Western blots show pull down of endogenous FIGNL1 following coIPs, *n* = 3 independent experiments. **c** 293 T cells were transfected with EV, FLAG-FIGNL1 or GFP-FLIP as indicated for 48 h. Cells were then treated with 2 µM cisplatin for 24 h prior to coIP, *n* = 2 independent experiments. **d** 293 T cells were transfected with FIGNL1-GFP together with either full-

length FLIP-FLAG or the indicated truncation mutants for 48 h prior to coIP and blotting for the indicated proteins, *n* = 3 independent experiments. **e** 293 T cells were transfected with FLIP-GFP and either full-length FIGNL1-FLAG or the indicated truncation mutants for 48 h prior to coIP and blotting for the indicated proteins, *n* = 2 independent experiments. **f** Schematic showing the FIGNL1 and FLIP truncations and the coIP results. Mutants are drawn to scale. Source data are provided as a Source Data file.

increased γH2AX staining (Fig. 7b), FLIP-OE cells failed to form RAD51 foci (Fig. 7b, c) in the absence of exogenous DNA damage. To further examine this, we treated control cells with cisplatin at doses that induced robust γH2AX and RAD51 foci formation (see Fig. 6e, f,

g). Compared to WT cells, FLIP-OE cells failed to form RAD51 foci even upon cisplatin treatment (Supplementary Fig. 7A, B) demonstrating either defective foci formation or, more likely, dissociation of RAD51 from foci.

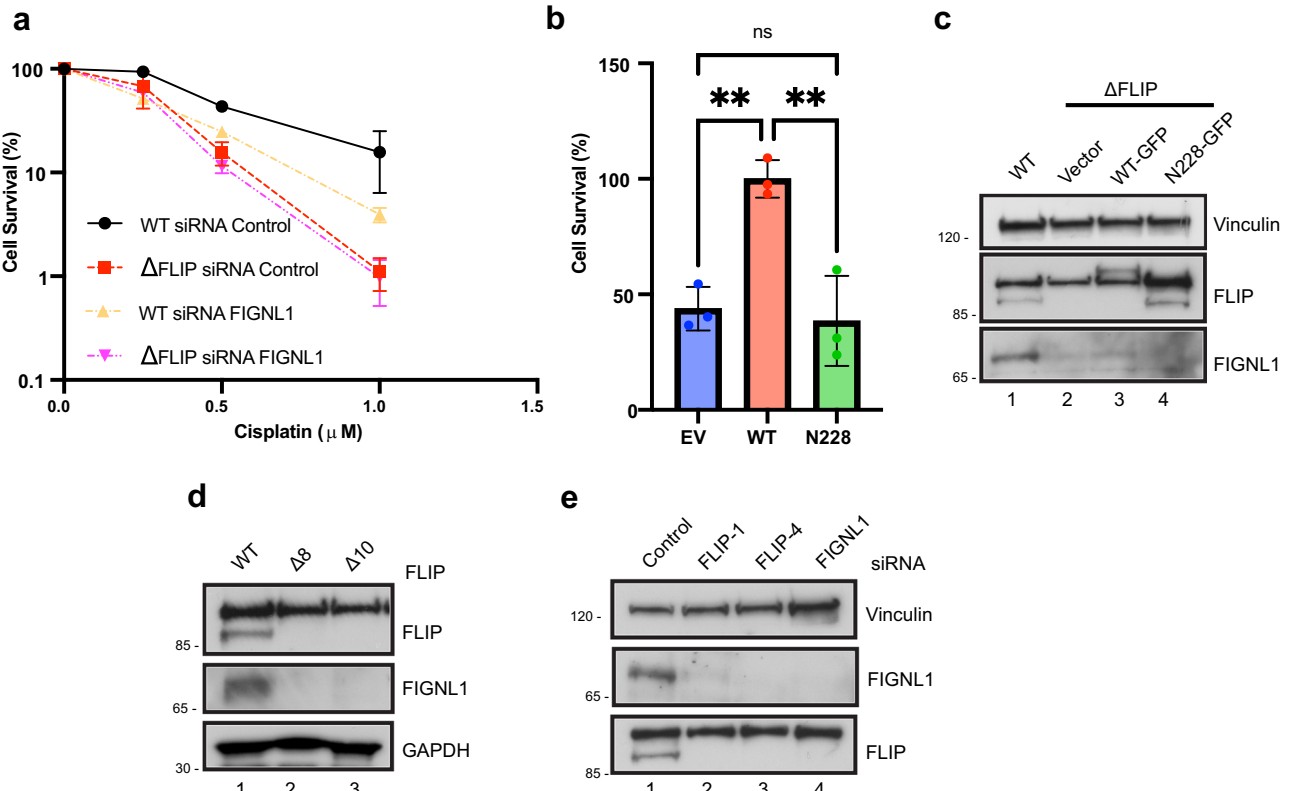

**Fig. 5 | FLIP and FIGNL1 function in the same pathway and form a stable complex. a** U2OS cells expressing control gRNA (WT) or FLIP null cells were treated with control or FIGNL1 siRNA for 48 h. Cells were then exposed to the indicated doses of cisplatin and processed for CSAs as above. *N* = 3 independent experiments, mean ± SEM shown. **b** ΔFLIP U2OS cells stably expressing EV, full length FLIP-FLAG (WT) or the N228 FLIP truncation mutant were exposed to 1 µM cisplatin for 24 h. Cells were then processed for CSAs as above. *N* = 3 independent experiments, mean ± SD shown, ns – not significant (*P* = 0.8795), **\*\*P* ≤ 0.01, one-way ANOVA followed by Tukey's. **c** Expression of FLIP and FIGNL1 in WT or the ΔFLIP rescue U2OS cell lines in (**b**). **d** WCLs from U2OS cells expressing control gRNA (WT) or two independent ΔFLIP clones were assayed for the indicated proteins by WB, *n* = 3 independent experiments. **e** U2OS cells were reverse transfected with the indicated siRNAs for 48 h. WCLs were prepared and WB were performed against the indicated proteins, *n* = 3 independent experiments. Source data are provided as a Source Data file.

SWSAP1 is a RAD51 paralog that is important for proper RAD51 foci formation upon DNA damage[50,51]. SWSAP1 was recently proposed to stabilize RAD51 filaments by counteracting FIGNL1, as FIGNL1 loss could promote restoration of RAD51 foci formation when SWSAP1 was depleted[49]. Since both proteins formed a complex, and to determine whether FLIP functioned in a similar manner, we depleted cells of SWSAP1 using siRNAs and co-transfected either control, FIGNL1 or FLIP siRNAs. Similar to what was previously shown for FIGNL1, we observed that loss of FLIP led to restoration of RAD51 foci formation in SWSAP1 depleted cells (Fig. 7d).

To further examine whether FLIP and FIGNL1 functioned in a similar manner, we examined whether FIGNL1 loss had additive or epistatic effects on the persistence of RAD51 foci on chromatin after cisplatin treatment (shown in Fig. 6). FIGNL1 depletion did not alter the kinetics of RAD51 foci persistence in the absence of FLIP (Fig. 7e), further buttressing their similar roles. Taken together, our data identifies FLIP and FIGNL1 as critical regulators of RAD51 chromatin association and removal from damage foci upon DNA damage.

### FLIP loss leads increased DNA damage and RAD51 foci formation in S phase

In addition to defective RAD51 removal from damage foci after exogenous DNA damage, we were intrigued by the marked increases in RAD51 foci formation in the absence of FLIP. Having observed increased chromosomal aberrations (Fig. 2b) and increased break

formation in the absence of FLIP (Fig. 3g), we reasoned that replication fork stalling and breakage may lead to accumulation of RAD51 during S phase, which then persists due to absence of FLIP. To test this, we labeled cells with EdU over a time course, detected EdU using click chemistry and co-stained for RAD51 foci formation. We reasoned that if RAD51 foci was arising in S phase, then there should be greater than expected overlap between EdU and RAD51 positive staining. EdU staining (Fig. 7f) progressively increased from 40% (1 hr) to around 70% (6 h) whereas RAD51 foci formation was unchanged at around 50% (Supplementary Fig. 7C). Remarkably, by 3 hr after EdU pulse, during which only 60% of the cells were stained, we observed nearly 90% overlap in RAD51 and EdU staining (Fig. 7g, h), as opposed to the expected 30% overlap, reaching nearly 100% by 6 hr staining (Fig. 7h). These data suggest that the increased RAD51 chromatin association and foci formation observed in the absence of FLIP (Fig. 6h, i) was likely due to problems arising during replication.

### FLIP/FIGNL1 promote replication fork progression

RAD51 association with DNA can drive fork reversal which might be expected to slow replication fork progression[29]. Our observation of increased DNA damage, break formation and increased RAD51 association with DNA during S phase upon FLIP loss in cells without exogenous DNA damage prompted us to determine whether FLIP loss affected replication fork progression. We sequentially pulsed cells with the DNA analogs CldU and IdU and, using single-molecule DNA fiber assays, examined fiber lengths of the second signal upon FLIP

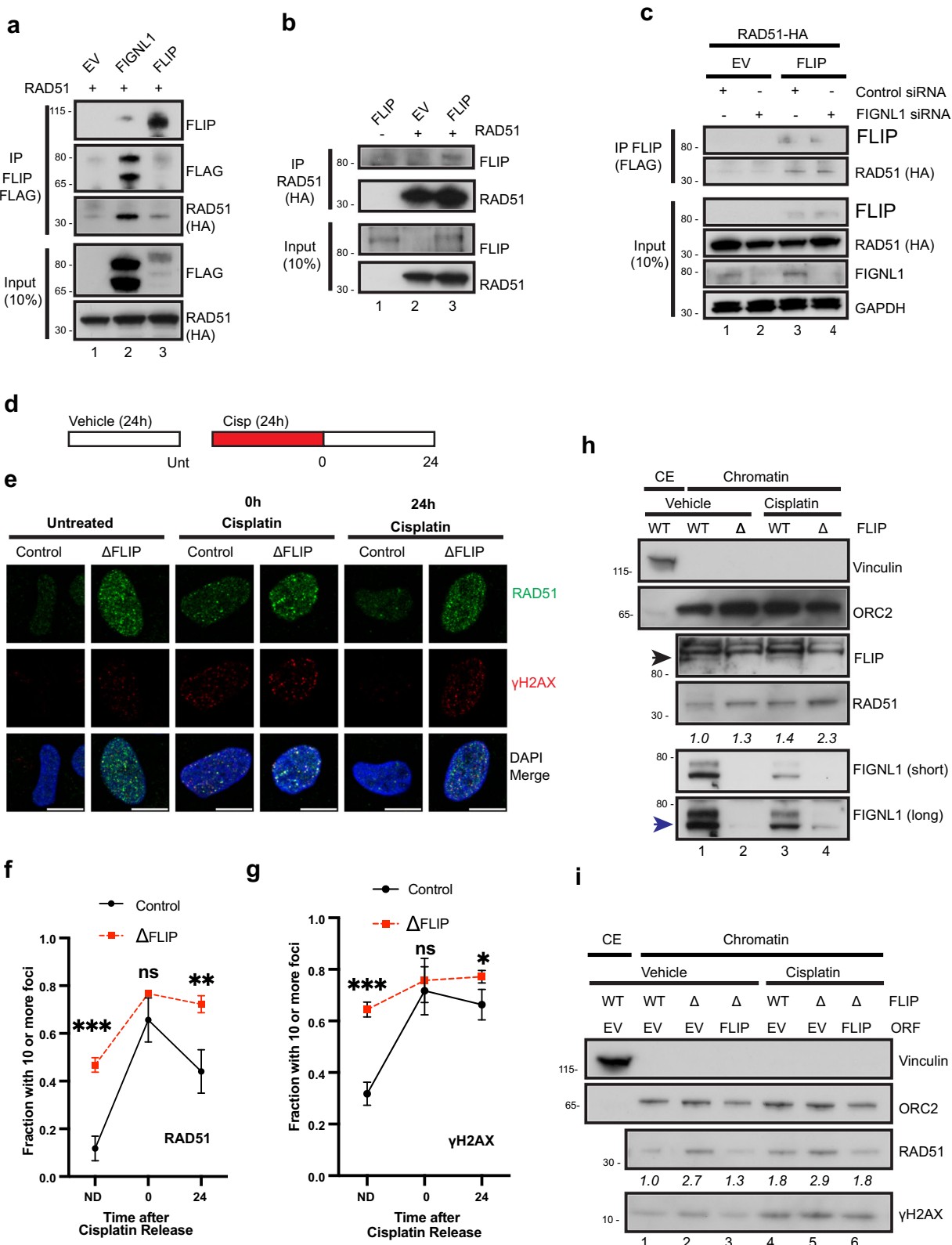

depletion. As a positive control we treated cells with the DNA polymerase alpha inhibitor CD437, whose inhibition was recently shown to slow both leading and lagging strand progression[52]. We found that FLIP loss led to significant reduction in the length of the IdU signal, consistent with slowing of replication fork progression in the absence of FLIP (Fig. 8a, b). Similar results were seen when FLIP was depleted by

siRNA as was seen in ΔFLIP cells. FIGNL1 depletion also led to similar reduction in fork progression as was seen upon FLIP loss (Fig. 8c). Unlike CD437 treatment, we did not observe significant increases in fork asymmetry, suggesting that FLIP loss might not in itself lead to uncoupling of the leading and lagging strands (Supplementary Fig. 8A–D).

**Fig. 6 | FLIP limits RAD51 chromatin association and persistence at damage foci. a** 293 T cells were transfected with RAD51-HA and either empty vector (EV), or the indicated 3X-FLAG-tagged proteins for 48 h prior to coIP using FLAG beads. Lysates were digested with benzonase prior to coIP, $n = 4$ independent experiments. **b** 293 T cells were transfected with EV, FLAG-tagged FLIP or HA-tagged RAD51 as indicated for 48 h prior to coIP using HA beads. Lysates were digested with benzonase prior to coIP, $n = 3$ independent experiments. **c** 293 T cells were reverse transfected with the indicated siRNAs. The next day, cells were transfected with EV, FLAG-tagged FLIP or HA-tagged RAD51 as indicated for 48 h prior to coIP using FLAG beads, $n = 2$ independent experiments. **d** Schematic of the experiment in (**e, f, g**). Cisplatin was added at 0.5 μM. **e** Control gRNA expressing and ΔFLIP U2OS cells were processed as shown in (**d**). Representative IF images show RAD51 and γH2AX foci. **f, g** Quantification of the experiment in (**d, e**). $N = 3$

independent experiments, mean ± SD shown. ND – no drug. Statistics represent unpaired $t$ tests (two-tailed) for each time point, ns – not significant; *$P = 0.0436$; **$P = 0.0075$; ***$P \leq 0.001$. **h** Control gRNA-expressing (WT) or ΔFLIP U2OS cells were treated with vehicle or 2.5 μM cisplatin for 18 h then fractionated prior to blotting for the indicated proteins. CE – cytoplasmic extract. Black arrowhead – FLIP. Blue arrowhead – FIGNL1, $n = 3$ independent experiments. **i** Control gRNA-expressing (WT) or ΔFLIP U2OS cells stably expressing empty vector (EV) or GFP-FLIP (FLIP) were treated with vehicle or 1.5 μM cisplatin for 18 h. Samples were processed for chromatin fraction prior to blotting for the indicated proteins, $n = 2$ independent experiments. CE – cytoplasmic extract. RAD51 numbers are quantified related to ORC2 loading control. Scale bars = 10 μm. Source data are provided as a Source Data file.

## FLIP/FIGNL1 is important for proper homologous recombination and sensitizes BRCA mutant cells to PARP inhibitor treatment

Having observed interaction between FLIP and RAD51, and altered RAD51 dynamics in the absence of FLIP, we sought to determine whether FLIP loss regulated HR. FIGNL1, which phenocopies FLIP loss with respect to RAD51 chromatin levels, has also been linked to reduced HR competency upon double stranded break (DSB) formation[32]. Accumulation of RAD51 on chromatin in the absence of FLIP might suggest anti-recombinase properties for FLIP, however, the requirement for FLIP following cisplatin treatment (Fig. 1) suggests that FLIP is important for proper completion of HR and that increased persistence of RAD51 in the absence of FLIP/FIGNL1 was likely due to a failure of the latter steps of HR. Consistent with a requirement for FLIP for proper HR, MCA assays showed that FLIP knockdown led to reduced viability after camptothecin (CPT) treatment (Fig. 8d), which causes replication-associated breaks that are repaired by HR[53]. However, since CPT also induces formation of R-loops that can lead to fork collapse[54], this phenotype may not necessarily reflect HR defects.

To directly examine HR competency at DSBs in the absence of FLIP, we took advantage of U2OS cells expressing a previously reported modified GFP reporter that fluoresces green when an I-SceI enzyme-induced DSB is repaired[55]. Using this assay, siRNA depletion of FLIP led to significant HR reduction (Fig. 8e). To further characterize a role for FLIP in regulating HR, we examined sensitivity to PARP inhibitor treatment. PARP inhibitors are synthetic lethal with loss of HR factors and have recently been approved to treat cancers with BRCA1/2 mutations[56–58]. To study modulation of PARP inhibitor sensitivity by FLIP or FIGNL1 loss, we made use of SUM149 cells breast cancer cells, which have mutations in the BRCA1 gene[59]. APEX2 knockdown, which we previously showed led to increased sensitivity to PARP inhibitor treatment in these cells[60], served as positive control. Here we show that FLIP or FIGNL1 knockdown significantly sensitized SUM149 cells to PARP inhibitor treatment (Fig. 8f). Taken together, our results demonstrate an HR function for FLIP and suggest that FLIP and FIGNL1 together promote viability after exposure to various kinds of DNA damage.

## Discussion

Here we have identified a novel protein, c1orf112, also known as FLIP / FIRRM, that plays vital roles in ICL repair and homologous recombination. Tight regulation of RAD51 levels is critical for proper recombination control. Our work identifies FLIP as a novel regulator of RAD51, determining its chromatin loading and foci formation both in the presence and absence of DNA damage. We envisage at least two roles for FLIP. During normal replication, FLIP regulates RAD51 chromatin association, allowing adequate fork progression and preventing increased break formation and chromosomal aberrations. Upon induction of exogenous DNA damage and perhaps subsequent to BRCA2-dependent RAD51 foci formation, FLIP regulates RAD51 dissociation from damage foci to promote completion of HR.

Although ΔFLIP cells showed exquisite sensitivity to ICL agents in particular, FLIP does not appear to be a Fanconi pathway gene. Indeed, FANC-ID activation and foci formation occurred normally in ΔFLIP cells, and although FLIP limits genomic instability after ICL treatment, we did not observe significant increases in characteristic radial formation often observed following loss of Fanconi genes. Recently, a Fanconi-independent and NEIL3-dependent sub-pathway of ICL repair was identified[41,61], whether FLIP functions downstream of both pathways is not clear. However, depletion of either FANCA or FLIP led to similar reduction in viability after ICLs, which was further increased upon loss of both proteins, pointing to FLIP acting in parallel to or, more likely, downstream of FANC-ID foci activation as might be expected since RAD51 nucleofilament formation and HR occurs late during ICL repair. Indeed, FANCD2 depletion prevented FLIP foci formation upon cisplatin treatment.

FLIP's role is closely tied to FIGNL1, with which it forms a stable complex. DepMap analysis revealed high correlation between loss of either gene, suggesting similar functions. We confirmed that both proteins interact and mapped the regions necessary for binding. FIGNL1 appears to bind to FLIP through its N-terminal 120 aa domain, which was both required and sufficient for binding to FLIP despite the reduced expression of this mutant compared to the WT. Interestingly, FIGNL1 mutants deficient in the N terminal 120 aa were previously shown to be defective in recruitment to damage sites and in promoting HR[32]. However, although this was consistent with the idea that FLIP recruits FIGNL1 to damage sites, the reduction in FIGNL1 levels upon FLIP loss and reduced FIGNL1 expression upon cisplatin treatment prevented us from directly determining this. Attempts at addressing this using a FIGNL1-GFP were also as of yet unsuccessful. FLIP binding to FIGNL1 similarly required the N-terminal region, and we demonstrated that mutants of FLIP that failed to bind to FIGNL1 could not rescue the sensitivity of ΔFLIP cells to cisplatin treatment.

Our current model for how FLIP regulates RAD51 is through stabilization and perhaps recruitment of FIGNL1, which likely performs the major effector function of controlling RAD51 levels on chromatin. Purified FIGNL1 promotes dissociation of RAD51 from ssDNA[49], providing a biochemical basis for our observed increased persistence of RAD51 in the absence of FLIP-FIGNL1 upon ICL treatment, and the dissociation of RAD51 from foci when FLIP is over-expressed. However, it is worth noting that while both proteins likely function in the same pathway, they do not have completely overlapping phenotypes. FIGNL1 depleted cells are not as sensitive to cisplatin as FLIP depleted cells are (Fig. 5a). FLIP can interact with RAD51 independently of FIGNL1 (Fig. 6c). FLIP overexpression is sufficient to prevent RAD51 foci formation independently of any perturbations to FIGNL1 levels (Fig. 7b, c and Supplementary Fig. 7A, B), and lastly, our FLIP-GFP rescue cell line does not fully restore FIGNL1 levels, yet significantly improves viability upon cisplatin treatment and RAD51 dissociation from chromatin (Figs. 5b, c, 6i). Nevertheless, in the majority of the phenotypes tested, there were similar outcomes. Indeed, multiple functions of FIGNL1 were conserved with FLIP such as the previously

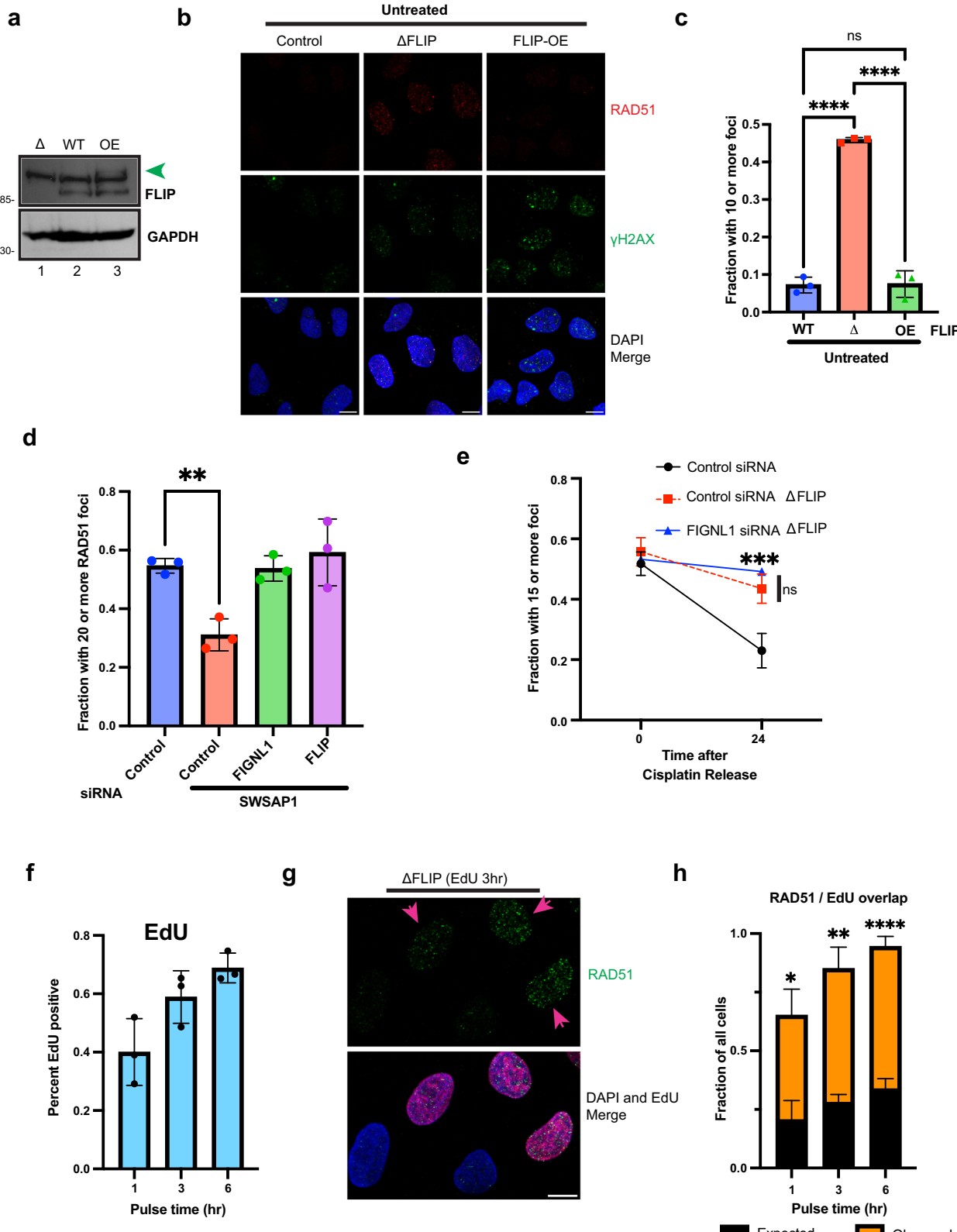

reported role of FIGNL1 in counteracting SWSAP1 activity in the regulation of RAD51 foci formation[49] and the failure of FIGNL1 depletion to further sensitize FLIP KO cells to cisplatin. This was to be expected due to the striking but incomplete (see Supplementary Fig. 5A) reduction in FIGNL1 protein levels upon loss of FLIP. Interestingly, loss of FIGNL1 also led to loss of FLIP expression, revealing that both proteins form a stable complex.

Our experiments showing striking sensitivity to ICL agents following FLIP-FIGNL1 loss and reduced but still significant sensitivity to HU or IR treatment is similar to what was seen with other RAD51 regulatory helicases ZGRF1 and HELQ[25,62–64]. These proteins, when absent, show defective HR using reporter assays and lead to increased persistence of RAD51 foci upon DNA damage. Although these all have HR functions, ICL repair appears to be particularly dependent on not just

**Fig. 7 | FLIP prevents increased RAD51 foci formation in S phase. a** Western blots show FLIP levels in knockout (Δ), U2OS (WT) and FLIP-GFP expressing (OE, green arrow) cells, *n* = 2 independent experiments. **b** Control, ΔFLIP or FLIP-GFP over-expressing (OE) U2OS cells were processed for IF. Representative IF images show RAD51 and γH2AX foci. **c** Quantification of (**b**), mean ± SD shown, *n* = 3 independent experiments, ordinary one-way ANOVA followed by Tukey's, ns – not significant (*P* = 0.9897), ****P* ≤ 0.0001. **d** U2OS cells were reverse transfected with control or the indicated siRNAs for 48 h. Cells were plated on coverslips, exposed to 100 nM camptothecin (CPT) for 20 h and processed for RAD51 IF, mean ± SD shown, *n* = 3 independent experiments. Ordinary one-way ANOVA followed by Dunnett's, ***P* = 0.0072. **e** U2OS cells expressing control sgRNA or FLIP KO U2OS cells were reverse transfected using the indicated siRNAs for 48 h, plated on coverslips, exposed to 0.5 μM cisplatin for 24 h and released for 0 h or 24 h prior to processing for RAD51 IF. Mean ± SD shown, *n* = 3 independent experiments, two-way ANOVA followed by Tukey's multiple comparison test, ns – not significant (*P* = 0.3567); ****P* ≤ 0.001. **f** Quantification of percent EdU positive cells after pulsing for the indicated durations, mean ± SD shown, *n* = 3 independent experiments. **g** FLIP KO U2OS cells were pulsed with EdU for 3 h. Cells were pre-extracted and stained for EdU (via Click-iT reaction) and RAD51. Pink arrows show RAD51 positive cells. Merged images show overlap between EdU staining (red) and RAD51 positive cells. **h** Plot showing observed overlap between RAD51 and EdU staining. Mean ± SD shown, *n* = 3 independent experiments. Expected values were obtained by multiplying EdU and RAD51 positive fractions. Statistics represent unpaired *t* tests (two-tailed) of expected vs observed values for each time point. **P* = 0.0109; ***P* = 0.0015; ****P* ≤ 0.0001. Scale bars = 10 μm. Source data are provided as a Source Data file.

RAD51's HR role but other functions linked to fork reversal and fork protection. In addition to FLIP, FIGNL1 also interacts with SPIDR[32] and SWSAP1[49]. SWS1, SWSAP1 and SPIDR together form a distinct complex, that was recently shown to be dispensable for intra-chromosomal HR but essential for inter-homolog HR[50], showing that various types of recombination might require multiple different regulators of RAD51 function. How FLIP regulates FIGNL1's multiple interactions will be a subject of future studies.

FLIP limits RAD51 levels on DNA even in the absence of DNA damage and, using EdU labeling, we showed that this likely occurred in S phase. A similar function has been attributed to RADX, which regulates the amounts of RAD51 at stalled forks[65–67] by competing with RAD51 for binding to ssDNA. Unlike RADX, FLIP and FIGNL1 are both required for proper HR consistent with a role downstream of RAD51 foci formation. Too much RAD51 on chromatin may be detrimental to replication as it might be necessary to finetune the fork reversal activities of RAD51 to prevent excessive and/or unwarranted fork reversal[68]. Failure to do so might lead to increased fork collapse. Consistent with this, we observed increased γH2AX upon FLIP loss. RAD51 can bind to both ssDNA and dsDNA, and its dsDNA binding activity has recently been proposed to be the basis for its role in fork protection[31]. It is not yet clear what the primary lesion of FLIP null cells culminating in the observed RAD51 foci formation in the absence of damage is; defective RAD51 dissociation from its normal process of repairing replication-associated breaks, or increased RAD51 chromatinization leading to excessive fork reversal and break formation. In any case, the end result is increased break formation and defective repair capacity.

Platinum based chemotherapeutics are widely used in the treatment of several cancer types. We show here that the FLIP-FIGNL1 complex is critical for viability after treatment with chemotherapeutic agents, suggesting that targeting this complex might be a potent combination approach to treatment of several cancers. PARP inhibitors have also recently been widely used for treatment of HR-deficient cancers[58,69]. We found that FLIP-FIGNL1 knockdown potentiates sensitivity to Olaparib treatment, likely because of FLIP-FIGNL1 roles in HR control. The ATPase dead mutant of FIGNL1 is still able to dissociate RAD51 from ssDNA in vitro and may not be an effective therapeutic approach[49], however, disruption of the interaction between FLIP and FIGNL1 using small peptides might be a potent strategy for treatment of various cancers.

## Methods

### Cell lines
All U2OS cells were passaged in McCoys 5A media supplemented with 10% fetal bovine serum and 1% penicillin/streptomycin. RPE1 cells were maintained in DMEM:F12 media supplemented with 10% FBS and 1% penicillin/streptomycin. HeLa cells and 293 T cells were grown in DMEM supplemented with 10% fetal bovine serum and 1% penicillin/streptomycin. SUM149 cells were grown in Ham's F12 media supplemented with 10% FBS and 1% penicillin/streptomycin. DR-GFP U2OS cells were previously described[70]. All cells were maintained at 37 °C in 5% CO$_2$, and passaged using 0.25% Trypsin-EDTA to dissociate cells.

### Plasmids and cloning
Myc-DDK-tagged lenti ORF clone of c1orf112 (Lenti-Myc-FLAG-FLIP) was obtained from Origene (RC211444L1). Human FIGNL1 sequence-verified cDNA (FIGNL1-cDNA) was obtained from Horizon Discovery (MHS6278-202759761). pcDNA3.1-3xFLAG was generated by inserting a DNA fragment coding 3xFLAG tag into the multiple cloning site of pcDNA3.1 using restriction enzymes Xba1 and Apa1. To make 3XFlag-FLIP or 3XFLAG-FIGNL1, each gene was amplified by PCR and cloned into pcDNA3.1-3XFLAG using KpnI and XhoI restriction enzymes sites. N-terminal HA-tagged RAD51 was generated by inserting an HA-RAD51 DNA fragment synthetized as a gBlock fragment by IDT into pcDNA3.1 vector using BamHI and ApaI sites. The FLIP N and C terminal truncations and FIGNL1 truncations were all obtained by PCR cloning. GFP-tagged-FLIP was obtained by restriction digestion from Lenti-Myc-FLAG-FLIP using AsiSI and MluI and cloning into a pCMV6-AC-GFP vector originally obtained from Origene. gRNA-target-resistant constructs were generated by site directed mutagenesis using Quik-Change II Site-directed mutagenesis kit (200521, Agilent) or Q5 Site-directed mutagenesis (E0554, NEB) according to the manufacturer's instructions. All primers used in this study are provided in Supplementary Data 2. All constructs used in this study were verified either by Sanger sequencing or whole plasmid sequencing by Primordium labs.

### Plasmid transfections and virus production/viral transduction
Plasmids were transfected into cells 293 T cells using PolyJet (SL100688, Fisher) according to the manufacturer's instructions. Cells were left untreated or treated with drugs and harvested 2–4 days later. To make stable cell lines, restriction-digest linearized plasmids were transfected into U2OS or HeLa cells using Lipofectamine 3000 reagent according to the manufacturer's instructions. 24 h later cells were selected using Geneticin for 5–7 days. With GFP-tagged constructs, selected cells were subsequently sorted using BD FACSymphony S6 (BD Biosciences). Lentiviruses were generated as previously described[33] in 293 T cells, filtered and used to infect target cells.

### siRNA transfections
RPE1, U2OS, 293 T or HeLa cells were reverse transfected into cells at 20–40 nM using Lipofectamine RNAiMAX reagent (13-778-075, Invitrogen) according to the manufacturer's instructions. Occasionally, seeded cells were transfected a second time with siRNAs the next day. Cells were then processed as described and harvested 2-4 days later. Unless otherwise indicated siRNAs used in this study were SmartPools from Dharmacon. Control siRNAs used were Allstars Negative control siRNAs (Qiagen).

### Generation of CRISPR-knockout FLIP cells
gRNA sequences were annealed and cloned into LentiCrispr v2 (Addgene) modified to express NAT resistance gene. After virus

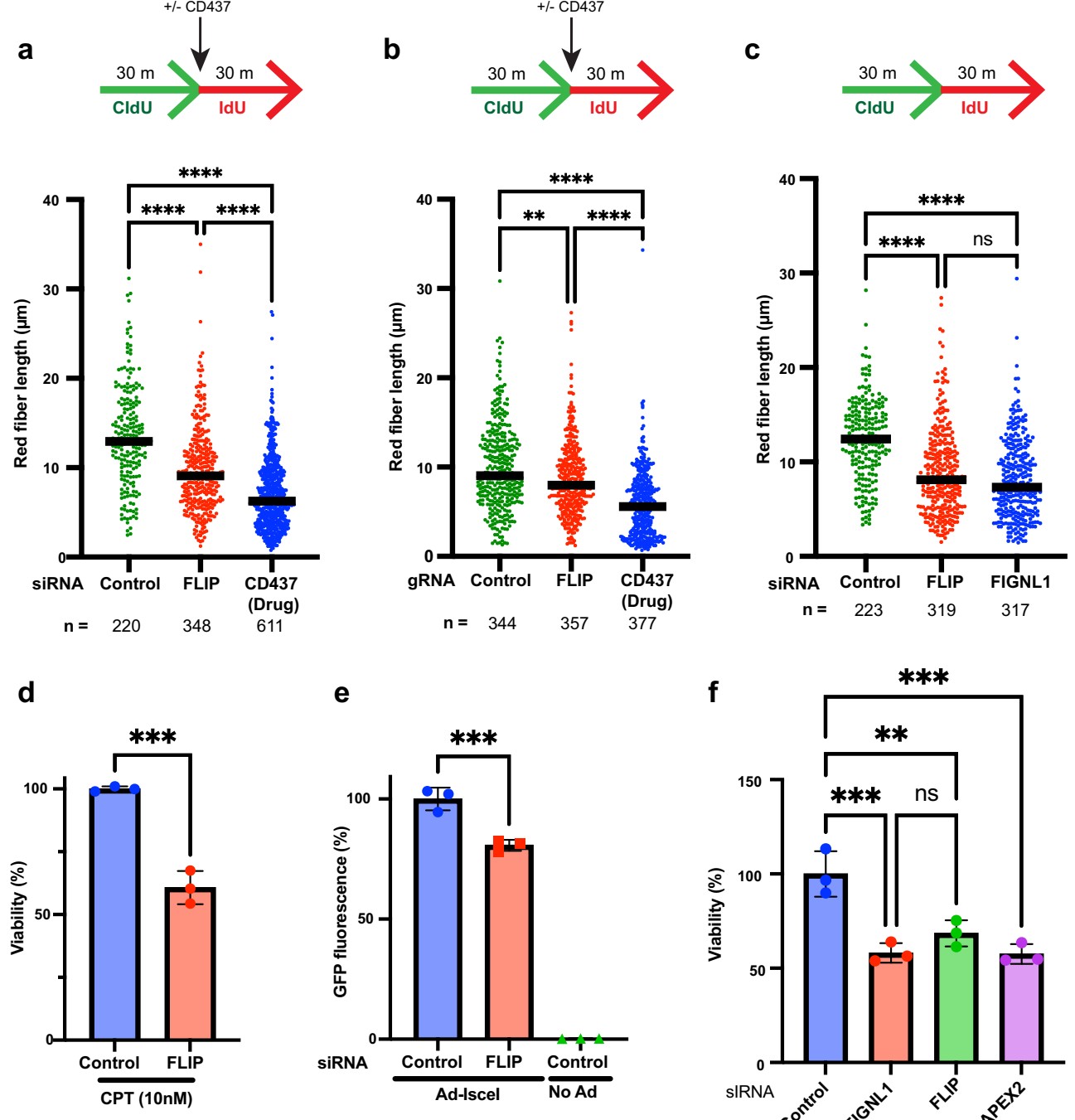

**Fig. 8 | FLIP / FIGNL1 promote replication fork progression and HR. a** Schematic of pulse-labeling DNA strands, and red fiber length of two-color DNA fibers. DNA fibers were from U2OS cells transfected with siRNAs as indicated or treated with CD437. **b** Schematic of pulse-labeling DNA strands, and red fiber length of two-color DNA fibers. DNA fibers were from U2OS cells expressing control gRNA, FLIP gRNA or control gRNA with 5 μM CD437 treatment. **c** Schematic of pulse-labeling DNA strands, and red fiber length of two-color DNA fibers. DNA fibers were from U2OS cells transfected with siRNAs as indicated. **a–c** Black line represents median, three independent experiments, representative experiment shown. Kruskal-Wallis followed by Dunn's, ns – not significant ($P = 0.0989$); **$P = 0.0032$; ****$P \le 0.0001$. **d** U2OS-GFP cells were transfected with control or FLIP siRNA. MCA showing survival of U2OS-GFP cells after treatment with the indicated dose of CPT for 24 h. Data is normalized to viability of control siRNA. Mean ± SD shown, $n = 3$ independent experiments, unpaired $t$ tests (two-tailed), ***$P = 0.0005$. **e** U2OS DR-GFP reporter cells were reverse transfected with the indicated siRNAs for 48 h and then infected or not with recombinant Adenovirus expressing I-SceI to cause a DSB within the reporter. 48 h later, GFP-positive cells, indicating HR efficiency, were detected by flow cytometry. Data is normalized to viability of control siRNA. Mean ± SD shown, $n = 3$ independent experiments, ordinary one-way ANOVA followed by Dunnet's, ***$P = 0.0004$. **f** SUM149 cells were reversed transfected with the indicated siRNAs for 48 h prior to treatment with the indicated dose of Olaparib for 24 h. After 2-3 days cellular viability was assayed by CellTiterGlo. Data is normalized to viability of control siRNA. $n = 3$ independent experiments, mean ± SD shown. Ordinary one-way ANOVA followed by Tukey's, ns – not significant ($P = 0.4195$); **$P = 0.0052$; ***$P \le 0.001$. Source data are provided as a Source Data file.

generation, U2OS cells were infected and selected using 250 µg/mL of NAT for 5 days. Single cells were cloned and FLIP knockout status was ascertained by western blotting. To confirm knockout status, genomic DNA was isolated using GeneJet Genomic DNA Purification Kit (K0702, ThermoFisher Scientific). gRNA target knockout regions were amplified by PCR and Sanger sequenced.

## Reagents

DMSO was obtained from Fisher Scientific (Cat. # 97063-136). Cisplatin was purchased from Selleck Chemicals (Cat. # S1166). Hydroxyurea was obtained from Sigma (Cat. # H8627) and dissolved in PBS. CD437 was obtained from Tocris Bioscience (Cat. # 1549). Olaparib was from Selleck Chemicals (Cat. # S1060). MMC was obtained from Santa Cruz Biotechnology (Cat. # sc-3514A). Puromycin from Sigma (Cat. # P8833). Geneticin from Gibco (Cat. # 0131035). Nourseothricin (NAT) Gold Biotechnology (Cat. # 501532818). Halt Protease and Phosphatase Inhibitor Cocktail was obtained from Fisher Biotech (Cat. # 78440).

## Immunoblots

Western blotting was done as previously described[33]. Briefly, cells were harvested and lysed in RIPA buffer on ice for 15–30 min, clarified and transferred to new tubes. Alternatively, for whole cell lysates, cells were harvested and lysed in SDS lysis buffer (50 mM Tris HCl pH 6.8, 100 mM NaCl, 1% SDS, 10 mM NaF, 7% glycerol) prior to sonication. Protein content was measured on an Eppendorf Biophotometer using Bradford reagent, sample buffer was added, and samples were analyzed using sodium dodecyl sulfate gel electrophoresis (SDS-PAGE). Membranes were blocked in 5% (wt/vol) milk in Tris-buffered saline with Tween (TBST) buffer and then probed with antibodies (see antibodies). Westerns were quantified using ImageJ.

## Antibodies

The antibodies used in this work are as follows: Anti-HA 1:1000 (Sigma, H3663-200), c1orf112 1:1000 (Sigma, HPA023778), FANCD2 F17 1:200 (Santa Cruz Biotechnology, sc-20022), FANCA 1:1000 (Bethyl, A301-980A), tubulin 1:1000 (Sigma, T4026-.2 ML), FANCI 1:1000 (Bethyl, A301-254A), Vinculin 1:1000 (Sigma, V9131-.2 ML), ATM 1:1000 (Abcam, ab81292), γH2AX (JBW301) 1:1000 (Sigma, 05-636), mouse M2 anti-FLAG 1:1000 (Sigma, F1804-200UG), Rabbit anti-GFP antibody 1:1000 (Abcam, ab6556), RPA32-P-S4/8 1:1000 (Bethyl, A300-245A), RPA32 9H8 1:200 (Santa Cruz Biotechnology, sc-56770), CHK1-P-S317 1:1000 (Cell Signaling, 2344 S), CHK1 G4 1:200 (Santa Cruz Biotechnology, sc-8408), FIGNL1 1:1000 (Proteintech, 17604-1-AP-150UL), GAPDH 1:8000 (Santa Cruz Biotechnology, sc-47724), ORC2 1:1000 (Abcam, ab68348), RAD51 1:1000 (Abcam, ab63801), Mouse monoclonal RAD51 1:1000 (Millipore-Sigma, 05-530-I), Rad51 Antibody (H-92) 1:1000 (Santa Cruz Biotechnology, sc-8349), Goat anti-Rabbit IgG (H + L) Highly Cross-Adsorbed Secondary Antibody Alexa Fluor Plus 488 1:400 (Invitrogen, A32731), Goat anti-Mouse IgG (H + L) Highly Cross-Adsorbed Secondary Antibody Alexa Fluor Plus 488 1:400 (Invitrogen, A32723), Goat anti-Rat IgG (H + L) Highly Cross-Adsorbed Secondary Antibody, Alexa Fluor Plus 488 1:400 (Invitrogen, A48262), Goat anti-Mouse IgG (H + L) Highly Cross-Adsorbed Secondary Antibody Alexa Fluor Plus 594 1:400 (Invitrogen, A32742), Goat anti-Rabbit IgG (H + L) Highly Cross-Adsorbed Secondary Antibody Alexa Fluor Plus 594 1:400 (Invitrogen, A32740).

## Co-IP analyses

In total, 293 T cells were transfected with the indicated plasmids for 48–60 h in 60 mm dishes. Occasionally, samples were treated with cisplatin or vehicle for 18–24 h. Cells were harvested in cold PBS and lysed in RIPA buffer without SDS with 500 units benzonase added in some cases. Lysates were clarified by centrifugation and the supernatants were precleared with Pierce™ Protein A/G Magnetic Beads (88802, Thermofisher) and immunoprecipitated using Anti-FLAG® M2 Magnetic Beads (M8823, Sigma) for 4 h at 4 °C. Beads were washed three times and boiled in sample buffer prior to western blotting.

## Chromatin fractionation

WT and/or FLIP KO U2OS cells previously transfected or not with the indicated siRNAs were seeded in 10 cm plates, treated with drugs as indicated, harvested and washed with cold PBS. Sedimented cells were resuspended in cold Solution 1 consisting of 10 mM Hepes (pH 7.9), 10 mM KCl, 1.5 mM MgCl₂, 0.34 M sucrose, 1 mM DTT, protease and phosphatase inhibitor cocktails. Triton X-100 was added to 0.1%. After a 5-min incubation, samples were centrifuged at 1300 × g for 5 min, and the supernatant removed as the soluble fraction. Sedimented nuclei were washed once with Solution 1 and lysed in Solution 2 (3 mM EDTA, 0.2 mM EGTA, 1 mM DTT, protease and phosphatase inhibitor cocktails) for 30 min on ice. Samples were centrifuged at 1300 × g for 5 min, and the chromatin-enriched pellets washed once with Solution 2 followed by resuspension in lysis buffer containing 50 mM Tris pH 6.8, 100 mM NaCl, 1.7% SDS, 7% glycerol and protease and phosphatase inhibitors. After sonication, protein content was measured. Sample buffer was added to 1X and the samples were boiled for western blotting.

## Comet assay

Comet assay was implemented using CometAssay Kit (R&D SYSTEMS, 4250-050-K) following the manual. Briefly, U2OS cells were harvested, washed and resuspended in ice-cold PBS. 10 µL of the cell suspension was mixed with 100 µL of melted agarose gel, and 50 µL of the mixture was spread onto a microscope glass slide. After the agarose gel was solidified at 4 °C, the slide was immersed in lysis solution for 60 min at 4 °C followed by a 30 min incubation in neutral electrophoresis buffer. Then, the slide was placed in an electric field for 75 min. After the electrophoresis, the slide was immersed in DNA precipitation solution for 30 min followed by a 30 min incubation in 70% ethanol. Once the slide was completely dried, it was stained with SYBR Gold (Invitrogen, S11494) to detect DNA of each nucleus, called comets. The images of comets were obtained with TissueFAXS (TissueGnostics), and they were analyzed manually with ImageJ. The percentage of the comet tail signal to the whole comet signal was calculated.

## Immunofluorescence assays

U2OS or HeLa cells were transfected or not with the indicated siRNAs, then 24 h later plated onto glass coverslips in 6-well plates. After 24–36 h, drug treatments were performed for the indicated durations. Cells were then washed with PBS, fixed with 4% paraformaldehyde for 15 min and extracted with 0.5% Triton X-100 in PBS for 10 min. Alternatively, cells were pre-extracted using cytoskeleton buffer (containing 10 mM piperazine-N,N'-bis(2-ethanesulfonic acid) (PIPES), pH 6.8, 100 mM NaCl, 300 mM sucrose, 1 mM MgCl₂, 1 mM EGTA as well as 0.5% Triton X-100, protease and phosphatase inhibitors) for 5 min on ice prior to the fixation and permeabilization steps. Cells were then blocked in 3% BSA in PBS and incubated with primary and secondary antibodies. The coverslips were mounted, and nuclei were visualized with DAPI Fluoromount-G (Southern Biotech, OB010020) and images were acquired using an LSM 780 NLO (Zeiss).

## Micronuclei quantification

WT U2OS cells were reverse transfected with control or FLIP siRNAs for 48 h. Cells were then seeded onto coverslips in 6-well dishes and treated with 0.5 µM cisplatin or vehicle for 24 h. Cisplatin treated cells were allowed to recover for an additional 24 h. Cells were fixed, stained with DAPI and processed for microscopy as above. Images were analyzed in Fiji (ImageJ) and micronuclei were quantified.

## EdU Click-iT assay

EdU Click-chemistry was performed according to the manufacturer's instructions (Sigma). Cells were labeled with a final concentration of 10 μM EdU for the indicated durations. Cells were pre-extracted and click chemistry was performed prior to co-staining with antibodies to RAD51 according to the IF protocol above.

## Multicolor competition assay (MCA)

GFP-labeled U2OS cells were reverse transfected with the indicated siRNAs at 20 nM using Lipofectamine RNAiMAX reagent (Invitrogen) while RFP-labeled U2OS cells were transfected with control siRNAs in the same way. The following day, transfection was repeated with the same siRNAs and transfection reagent to maximize the knockdown effect. GFP- and RFP-labeled cells were mixed in equal quantities in six-well plates 2 days after the first siRNA transfection, and they were treated with the indicated dose of drug or vehicle control for 24 h. Fresh media was added and cells were maintained for 6 days after treatment. Subsequently, the percentage of GFP and RFP labeled cells were quantified by FACS analyses using BD FACS Canto II. Data were analyzed using FlowJo software (version 10).

## Cell cycle analyses

Cell cycle analyses were performed as previously reported[33]. Briefly, siRNA treated WT U2OS and/or FLIP knockout cells were treated or not with drugs as indicated. Cells were then harvested, fixed in 4% paraformaldehyde for 15 min at room temperature, or cells were fixed in 70% ethanol for 15 min on ice. After pelleting, cells were washed in PBS, resuspended in 50 μg/ml propidium iodide solution containing 0.1 mg/ml RNase A as well as 0.05% Trition X-100 for 40 min at 37 °C, resuspended in PBS and flow cytometry was performed using BD FACSymphony A5 or BD FACS Canto II. Data were analyzed using Flowjo software.

## DNA fiber assay

U2OS cells were incubated with 25 μM CldU for 30 min, washed and subsequently treated with 250 μM IdU for 30 min. After labeling, cells were washed, harvested and resuspended in PBS. 2 μL of the cell suspension were transferred to a glass microscope slide, overlaid with 6 μL lysis buffer (0.5% SDS, 200 mM Tris-HCl (pH 7.4), and 50 mM EDTA), and the slide was tilted to allow DNA to spread by gravity. After air-drying, 3:1 methanol/acetic acid was applied on the slides to fix DNA. DNA was denatured by incubating the slide in 2.5 M HCl for 80 min, followed by wash with PBS. Blocking was performed with 5% BSA in PBS for 30 min. For immunostaining, slides were incubated overnight with primary antibodies; ab6326 anti-BrdU (cross-reacts with CldU) antibody (rat) (1:100) and BD Biosciences 347580 anti-BrdU (cross-reacts with IdU) antibody (mouse) (1:25). Slides were washed with PBS followed by incubation for one hour with the secondary antibodies; anti-rat Alexa-488 antibody (1:400) and anti-mouse Alexa-594 antibody (1:400). After wash with PBS, mounting medium was added on the slides and images were acquired with Leica SP8 confocal microscope. Images were analyzed with ImageJ.

## Metaphase spreads

HeLa cells were reverse transfected with the indicated siRNAs at 20 nM using Lipofectamine RNAiMAX reagent (Invitrogen). The following day, transfection was repeated with the same siRNAs and transfection reagent to maximize the knockdown effect. Two days after from the first transfection, cells were treated with 2 ng/ml MMC or PBS for 48 h. Following treatment, the cells were exposed to colcemid (100 ng/ml) for 2 h, treated with a hypotonic solution (1:2 mixture of 0.075 M KCl and 0.9% Na citrate) for 10 min and fixed with 3:1 methanol: acetic acid. Slides were stained with Giemsa stain and 50 metaphase spreads were scored for aberrations. Metaphase spreads were observed using a Zeiss Axiovert 200 M microscope and captured using AxioVision.

## Clonogenic survival assay

Colony formation assays were performed as previously described[33]. Briefly, where indicated, cells were reverse transfected with siRNA to the indicated genes as described above. Afterwards, WT or CRISPR KO cells were exposed to the indicated doses of drugs for 16–24 h and adjusted for plating depending on dose of drug in six-well plates. After 7-10 days, cells were fixed and stained using crystal violet and scored with a colony counting pen (VWR).

## Cell viability assays

Cell viability assays using CellTiter-Glo 2 (Promega) were performed according to the manufacturers' instructions. Briefly, following siRNA treatment, cells were seeded at 500 to 1000 cells per well in triplicate in 96 well plates. Cells were then treated with the indicated drugs for 24 h, washed and left to recover for 48–72 h prior to being read on a BioTek FLx800.

## Homologous recombination assay

U2OS cells with a stably integrated DR-GFP reporter[71] were transfected with the indicated siRNAs 48 h prior to infection with adenovirus expressing I-SceI restriction enzyme at an MOI of 10. The percentage of GFP-positive cells was determined 48 h after infection by flow cytometry using a BD FACSymphony A5 (Becton Dickinson). Data were collected using BD FACS Diva software (v9, Becton Dickinson), and analysis was performed using FlowJo Software.

## Quantification and statistical analyses

Statistical analyses were performed using Prism 9 (GraphPad). All statistics used are indicated in the figure legends. For all experiments: n.s. $P \geq 0.05$; $*P < 0.05$; $**P < 0.01$; $***P < 0.001$; $****P < 0.0001$. All experiments in this work were performed at least three independent times (except for specific cases mentioned in the figure legends). Multiple siRNAs, multiple knockout clones, and multiple cell lines were analyzed to confirm that results were not caused by off-target effects or clonal variations. Representative images and/or experiments are shown for westerns, cell cycle analyses, DNA fiber assays and IFs. All source data are provided in the source data file.

## Reporting summary

Further information on research design is available in the Nature Portfolio Reporting Summary linked to this article.

# Data availability

All data supporting the findings in this study are provided in the main manuscript and/or its Supplementary Information files. External data sources such as BioGrid and DepMap were used for data analyses. Source data are provided with this paper.

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

## Acknowledgements

We thank all the members of the Adeyemi Lab for helpful discussions. This work was supported by an Early Career Investigator Grant from the Ovarian Cancer Research Alliance (ECIG-2023-3-1004), the NIH NCI Cancer Center Support Grant Early Investigator Award (P30 CA015704) and an NIH NIGMS R35 GM150532-01 (to ROA). This research was also supported by the Cellular Imaging Shared Resource RRID:SCR_022609 and the Flow Cytometry Shared Resource, RRID:SCR_022613, of the Fred Hutch / University of Washington / Seattle Children's Cancer Consortium (P30 CA015704).

## Author contributions

Conceptualization: R.O.A., Methodology: J.D.T., H.T., R.B., R.O.A., Investigation: J.D.T., H.T., R.B., T.T.O., A.P., R.O.A., Visualization: J.D.T., H.T., R.B., R.O.A., Supervision: R.O.A., Writing—original draft: R.O.A., Writing—review & editing: J.D.T., H.T., R.B., R.O.A.

## Competing interests

The authors declare no competing interests.
