## [Peer Review File · Nature Communications]

FLIP(C1orf112)-FIGNL1 Complex Regulates RAD51 Chromatin Association to Promote Viability After Replication StressREVIEWER COMMENTS

Reviewer #1 (Remarks to the Author):

In the manuscript by Tischler et al, the authors describe RADIF, as a novel gene involved in crosslink repair that they named for its association with FIGNL1. The authors demonstrate that RADIF functions during crosslink repair and that its loss results in DNA damage sensitivity to cisplatin, replication defects, and to increased RAD51 and gamma-H2AX foci. This gene had previously been described as FLIP in plants and is more accurately named FLIP since RADIF is not radiation sensitive as the "RAD" name implies. Furthermore, the phenotypes observed with RADIF are hard to uncouple from FIGNL1 since FIGNL1 protein levels are intimately linked to RADIF presence. This conundrum is never addressed and is a major caveat of the work. While the paper is interesting and presents some intriguing findings about RADIF, the study would be strengthened from a broader understanding of why an "anti-recombinase" does not increase recombination when disrupted.

Major Comments:

1. The authors should strongly consider calling RADIF, FLIP. It was already named FLIP in plants and changing the nomenclature is very confusing. The FLIP name makes a lot more sense, "FIGNL1 interacting protein" whereas RADIF does not. RAD generally relates to radiation sensitivity and this gene does not result in radiation sensitivity, causing even more confusion.
2. The over expression data presented in Figure 3G was unconvincing. It was difficult to observe such small differences in the western blot shown and whether or not that level of potential signaling is biologically significant is unknown. The data should be repeated to show clearer results, removed, or the language softened significantly. "Taken together, the increased damage signaling observed both when RADIF was lost or overexpressed showed that alterations in RADIF levels was highly correlated with dysregulated DNA damage signaling, suggesting a need for tight regulation of cellular RADIF levels to prevent genomic instability."
3. In Figure 6, RADIF was shown to interact with RAD51. However, it is possible that this interaction is indirect and mediated through FIGNL1. Similarly, the increase in RAD51 foci may also be due to loss of FIGNL1 in the RADIF KO cells. Therefore, the sentence "These results suggest that RADIF regulates dissociation of RAD51 nucleofilaments upon DNA damage and provide a possible mechanism for DNA repair defects seen upon RADIF loss after ICL agent treatment." should be softened.
4. It is also important to note that RAD51 foci is not a direct measure of dissociation of RAD51 nucleofilaments or defective filament formation and this language should be removed throughout the text. An example is in Point 4 above and also in Figure 6 "Compared to WT cells, RADIF-OE cells failed to form RAD51 foci even upon cisplatin treatment (Figure 6H-I) demonstrating either defective nucleofilament formation or, more likely, dissociation of RAD51 from nucleofilaments. Taken together, our data identifies RADIF and FIGNL1 as critical regulators of RAD51 chromatin association and nucleofilament disassembly upon DNA damage."
5. It is confusing that FIGNL1-RADIF are antirecombinases and yet loss of RADIF does not lead to more recombination as observed in a drGFP assay. Can a gene be an antirecombinase if recombination is not increased? The findings here demonstrate that there are more chromatid breaks upon RADIF disruption. Perhaps, sister chromatid exchange experiments performed with MMC or cisplatin will reveal if the anti-recombination phenotype is context specific. For example, the cells are not IR sensitive and so perhaps FIGNL1-RADIF only functions as an antirecombinase in a replicative context that is missed when an endonuclease induced break is formed in the drGFP assay.

Minor Comments:

1. In Figure 1B-F, the timing of the experiment should be indicated for the different treatments.

2. The labeling in Figure 4 is confusing. It is hard to tell what is tagged with what from the figure itself. Were these experiments performed in the presence of DNase? Is Pre-IP, input? Input is a more typical way to refer to the samples.

3. Supplementary Figure 5F, the blot is a very poor quality, and it is hard to see if RADIF is being depleted. Was this experiment repeated more than once? A better blot should be included or the data should be removed.

4. This sentence is very speculative and doesn't belong in the results. "Taken together, our results demonstrate an HR function for RADIF and suggest that RADIF inhibition might be a viable way to enhance sensitivity of BRCA1 mutant cells to PARP inhibitor treatment.

Reviewer #2 (Remarks to the Author):

This manuscript characterizes the function of C1orf112 in inter-strand cross link (ICL) repair. The authors propose that C1orf112 regulates RAD51 in a manner dependent on its ability to interact with FIGNL1 thereby naming the protein RADIF (RAD51-regulatory Interactor of FIGNL1). RADIF was identified as one of the top scoring genes from 4 previously published datasets screened for cisplatin sensitivity. When RADIF is lost, cells exhibit hallmarks of extensive genome instability characterized by increases in micronuclei formation and chromosomal abnormalities. RADIF localizes to sites of damage but functions in parallel to FA pathway to repair ICLs. Using BioGrid interactome data, the authors investigate interaction of RADIF with FIGNL1 to show that N-terminal regions of both proteins help form a co-stable complex and that FIGNL1 functions like RADIF in repairing ICL lesions. Interestingly, FIGNL1 is downregulated in RADIF-proficient cells after cisplatin exposure. Further characterization reveals that loss of RADIF or FIGNL1 causes increased RAD51 association with chromatin. In cisplatin-treated cells, RAD51 persists post recovery suggesting that RADIF-FIGNL1 complex functions to dissociate RAD51 post nucleofilament assembly to mediate repair. RADIF inactivation causes defects in replication fork elongation and sensitizes BRCA1-deficient SUM149 cells to PARPi.

This is a potentially interesting manuscript describing RADIF function in ICL repair. The authors have pursued several avenues to support their conclusions. However, much of the data included requires further characterization and reproducibility. Specifically, the mechanism by which RAD51 is disassembled is not thoroughly elucidated. Which RAD51-DNA intermediate does RADIF act on? Why is replication fork integrity compromised when RADIF is lost? Unlike RADIF, the genetic dependencies shown for FIGNL1 is limited and requires additional work, using the described truncation mutants, to support the claim that FIGNL1 and RADIF function together to regulate RAD51 in ICL repair.

Major comments:

The authors characterized the interaction between RADIF and FIGNL1 but it's not clear whether FIGNL1 was also identified as a hit from the genetic screens. If FIGNL1/RADIF complex is important for cisplatin repair, why was FIGNL1 not identified? FIGNL1 is not seen in the list. What was the confidence score compared to RADIF? Also, parallel characterization of FIGNL1 is needed to support conclusions derived for RADIF function in the context of results presented in Figure 7. Are RADIF and FIGNL1 functions always co-dependent?

Whether the interaction between RADIF and FIGNL1 is sufficient to promote viability to ICL is not thoroughly investigated. Although N228 mutant is unable to rescue viability, it can mean N228 is defective in additional interactions. Authors need to explore RADIF interactome and show changes in other interactions cannot explain the inability of the N228 construct to rescue.

Figure 6A results are not conclusive. RADIF expression seems variable across experiments (compare to Fig 4A where expression is robust). Authors should repeat and test for robust pulldown of RAD51. RAD51 pulldown also seems comparable to EV. A reverse IP is also necessary to confirm interaction. Are these interactions DNA dependent?

There is some mechanistic ambiguity in terms of how RAD51 is precisely regulated. Why are RAD51 foci higher in untreated RADIF-deficient cells? RADIF-depleted cells exhibit increased DNA damage which could be explained by increased DNA breaks but evidence for this result is not shown. Are the RAD51 foci in untreated conditions observed in all phases or is the increase specific to replication? Data suggesting that dissociation of RAD51 after DNA damage repair is defective is not thoroughly tested. The increase in chromatin bound RAD51 post 72 hours is reflective of inherently increased presence of RAD51 on chromatin. Experiments showing RAD51 recruitment and dissociation dynamics specifically at break sites is required to support the conclusion of a function in RAD51 dissociation.

The authors conclude in Figure 1H/I that RADIF depleted cells undergo increased apoptosis after cisplatin treatment but no evidence is provided to support this claim.

The authors claim that the doses of HU do not damage DNA. 2mM HU is generally considered high-dose that is sufficient to induce breaks within few hours.

In figure 3F, increased RPA phosphorylation is only observed in one clone. What is the reason for this discrepancy between the two clones? The reported RADIF clones seems to behave differently.

Without stable C-terminal truncation mutants for RADIF, it is not possible to conclude only the N-terminus region is involved in interaction with FIGNL1. The background for RADIF in the IP lane is also very high to confidently rule out lack of RADIF interaction with N120 FIGNL1 construct.

Is the fork slowdown dependent on increased RAD51 association with forks? Are similar effects (fig 6 and 7) observed in FIGNL1 depleted cells? Is RADIF recruited to replication forks to counteract RAD51?

Other comments:

The title of the manuscript mentions 'replication stress' is misleading given the authors observe sensitivity only to a subset of stress inducing agents (MMC, Cisplatin and CPT) and not HU.

Line 120: Which cell line was different? Is it relevant?

Line 122: It seems that 500 was randomly selected as a cut-off.

Line 127: Figure 1A: Are the genes listed in a ranking order? What is the combined confidence score for C1orf112 from the 3 screens? How 'high' is very highly? Is it comparable to known FANC genes for example? Numerical details should be included instead of mentioning 'very highly'.

Line 140: What are other forms of DNA damage?

Fig S1B, 1D,E,F: Statistical values are not included. RPE1 CSA data seems to include only 1 repeat (error bars are missing?)

Line 171: Indicate the low dose of MMS used

Figure S2A: Median is 0 for all conditions. How are the differences quantified to be statistically different?

Figure S2C: Which foci were quantified?

Figure S2D: The changes in ubiquitinated FANCD2 look very minimal.

Figure 3D: CtIP depletion has higher than 100% of RADIF foci. How is it possible?

Figure 3G: Is the overexpression 2-fold? Quantification to support this statement is not provided. How many times was this experiment repeated? Why did the authors generate 'near-endogenous levels of GFP-tagged RADIF' when the purpose was to test the effects of overexpression? The data seems inconsistent with the rationale.

Figure 4D: line 272: Statement that expression was 'better' is not supported with quantification. Over how many repeats was it observed? The expected molecular weight bands for the truncations should be indicated. Also include IF images to show localization was not affected for the stable proteins.

Line 284: Details are not provided as to how FIGNL1 participates in HR. Which HR function was disrupted?

Line 308: Survival is only reflected in cells treated with cisplatin and MMC suggesting it not generalizable to all kinds of replication stress.

Figure 5B: Corresponding western blot is not included. Does the RADIF N228 mutant localize to damage sites?

Figure 5C-F: Did the authors test whether FIGNL1 levels are restored when mutants of RADIF are re-expressed?

Figure 7D: Changes in DR-GFP signal are very minimal.

In figure 3C and 3D, the authors show that FANCD2 loss reduces RADIF foci formation but CtIP loss does not. Is foci formation disrupted in the absence of BRCA2?

Reviewer #3 (Remarks to the Author):

This manuscript describes the functional analysis of a poorly characterized human protein, termed RADIF, which was identified in previous genome-wide screens for factors that confer resistance to the DNA cross-linking agent cisplatin. The authors demonstrate that loss of RADIF leads to accumulation of DNA damage, increased chromosomal instability and altered cell cycle progression upon exposure of cells to cisplatin. In addition, cells lacking RADIF displayed inefficient DSB repair by homologous recombination (HR) and reduced replication fork velocity as compared to normal cells. Further analysis revealed that RADIF accumulates at nuclear foci in response to cisplatin treatment and forms a complex with FIGNL1, which functions as an anti-recombinase by dismantling RAD51 filaments. Moreover, RADIF was found to act in a common pathway with FIGNL1 to confer resistance to cisplatin, and its deficiency led to RAD51 accumulation on chromatin both in the absence and presence of exogenously induced DNA damage. Based on these results, the authors propose that RADIF is essential for limiting RAD51 levels on undamaged chromatin and for dissociation of RAD51 nucleofilaments to properly complete HR. Overall, the data presented are interesting, but most of the key experiments were performed only twice, and in some cases even only once, which casts doubt on the entire study. Moreover, further experiments are needed to gain mechanistic insights into the role of RADIF in RAD51 regulation.

Specific points:

1. Experiments must be repeated at least three times (clonogenic/viability assays, measurements of chromosomal aberrations/micronuclei, quantifications of nuclear foci, DR-GFP assays).
2. Based on the results of multicolor competition assays presented in Fig. 1B, the authors conclude that RADIF-depleted cells are hypersensitive to cisplatin, but not to hydroxyurea (HU) or ionizing radiation (IR). In the same experiment, they found that HU did not compromise the viability of FANCD2-depleted cells, which is not in agreement with clonogenic survival data in the literature (e.g. 10.1093/nar/gkx847). Therefore, sensitivity of RADIF-deficient cells to HU and IR should be also tested by clonogenic survival assay.
3. The authors state in the manuscript that RADIF is critical for ICL repair (e.g. page 4, line 104 or page 6, line 189). What is the evidence for this claim? It is possible that ICL agents induce cell death not only through the generation of DNA crosslinks (e.g. cisplatin or MMC increase the cellular levels of ROS). RADIF could be also involved in ICL traverse. In fact, the authors found that RADIF promotes cell survival after cisplatin treatment independently of the FA pathway that is implicated in ICL repair (Fig. 2D).
4. The statement "RADIF ... and promotes repair following DNA damage" (page 6, line 217) is not supported by the data presented in Fig. 3.
5. The western blot data in Fig. 3E-G showing the effect of RADIF depletion/overexpression on the cellular levels of DNA damage markers upon cisplatin treatment are not very convincing. For example, phospho-RPA is increased in delta8 clone but not in delta10 clone (compare lanes 4-6 in Fig. 3F). In any case, only minor differences are observed. Could the authors quantify IF images (QIBC analysis)?
6. Is the accumulation of RADIF foci after cisplatin treatment cell cycle dependent? Co-staining with PCNA antibody or EdU pulse-labeling could be performed to address this question.
7. Fig. 4: Interaction between RAD51 and FIGNL1 should be confirmed by co-immunoprecipitation assay of endogenous proteins. To test whether the interaction is direct, a pull-down assay with purified recombinant proteins could be performed. For Fig. 4C, reciprocal co-IP should be performed to confirm the negative effect of cisplatin treatment on FIGNL1-RADIF complex formation. For Fig. 4D, the blot of IPed RADIF variants should be added. For Fig. 4E, the presence of RADIF band in lane 3 (IP) is not clearly seen.
8. Fig. 5B: The lack of complementation of the cell survival defect of RADIF knockout cells with RADIF deltaN228 mutant does not necessarily mean that the defect is caused by loss of RADIF-FIGNL1 interaction. The deleted N-terminal region (large deletion, 227 aa) could also have another function(s) that is required for cell viability.
9. Based on the data shown in Fig. 5 C and D, the authors conclude that FIGNL1 is targeted for proteasomal degradation upon cisplatin treatment. However, the levels of ectopically expressed FIGNL1 are not affected by cisplatin treatment (Fig. 4C). Could the authors explain this discrepancy? If cisplatin treatment causes FIGNL1 degradation, how might RADIF and FIGNL1 function as a complex to prevent cisplatin-induced cell death?
10. Fig. 6A: co-IP data showing RADIF-RAD51 interaction are not very convincing (RAD51 pull-down with EV is also seen). Moreover, co-IP is done again only with ectopically expressed proteins.
11. The authors should examine whether RAD51 foci colocalize with RADIF foci following cisplatin treatment to further support the conclusion that RADIF physically interacts with RAD51 and regulates its levels on chromatin.
12. The authors conclude that RADIF acts in conjunction with FIGNL1 anti-recombinase to regulate RAD51 filament assembly. Cells lacking the RAD51 paralog SWSAP1 have been shown to exhibit defective RAD51 focus formation, which is suppressed by FIGNL1 depletion (DOI: 10.1038/s41467-019-09190-1). Does RADIF depletion also suppresses this phenotype of SWSAP1-depleted cells? In addition, the effect of FIGNL1 depletion on the level of RAD51 foci in RADIF KO cells should be evaluated to determine whether these proteins act in a common pathway. The results of these experiments can provide strong support for the above hypothesis.
13. The authors suggest that the observed reduction in replication fork velocity in RADIF-deficient cells might be a consequence of increased levels of RAD51 on chromatin, causing fork reversal. This hypothesis should be tested by experimentally, for example by depleting ZRANB3 translocase, which

mediates fork reversal. It will be also interesting to test whether replication fork progression is compromised by FIGNL1 depletion.

Other points:

1. Please show data points in the bar graphs (data from at least 3 independent experiments should be presented).
2. Statistical tests used should be described in figure legends.
3. The authors should not use the phrase "co-stable complex", which sounds strange.
4. Page 3, lines 78-82: Please insert citations.
5. Page 3, lines 91-93: Please insert citations.
6. Fig. 2D: the y-axis should be labeled "Cell Survival (%)". The same applies to Fig. 5A and 5B.
7. Page 27, line 982: "chromosomal instability" would be more appropriate here.
8. Fig. 3G: The blot should be probed with anti-GFP antibody to confirm expression of RADIF-GFP. The same applies to Fig. S5G.
9. Fig. 5D: Control without cisplatin should be added.
10. Page 10, line 406: "... promotes replication fork progression" is more appropriate here.
11. Page 11, lines 432-434: The authors state: "Consistent with a requirement for RADIF for proper HR, MCA assays showed that RADIF knockdown led to reduced viability after camptothecin (CPT) treatment ...". It should be noted that the sensitivity of cells to CPT does not necessarily reflect a defect in HR". CPT also induces formation of R-loops which can lead to replication fork collapse.
12. Page 15, line 617: Corrections needed.
13. Micronucleus assay should be described in Methods.
14. Fig. 7A and B: It should be indicated in the schematic when CD437 was added to cells. CD437 concentration used should be described in figure legends.
15. All figures showing IF images should have a scale bar.

RADIF response to reviewers

In the manuscript by Tischler et al, the authors describe RADIF, as a novel gene involved in crosslink repair that they named for its association with FIGNL1. The authors demonstrate that RADIF functions during crosslink repair and that its loss results in DNA damage sensitivity to cisplatin, replication defects, and to increased RAD51 and gamma-H2AX foci. This gene had previously been described as FLIP in plants and is more accurately named FLIP since RADIF is not radiation sensitive as the “RAD” name implies. Furthermore, the phenotypes observed with RADIF are hard to uncouple from FIGNL1 since FIGNL1 protein levels are intimately linked to RADIF presence. This conundrum is never addressed and is a major caveat of the work. While the paper is interesting and presents some intriguing findings about RADIF, the study would be strengthened from a broader understanding of why an “anti-recombinase” does not increase recombination when disrupted.

We thank the reviewer for the positive comments regarding our work. While we agree that making conclusions regarding RADIF/FLIP function is affected by concomitant disruption in FIGNL1 levels, we believe that our work showed certain unique FLIP phenotypes that allow for such conclusions. For example, we show that FLIP null cells were more sensitive to cisplatin than FIGNL1 depleted cells although there was epistasis when both proteins were knocked down. We also provided evidence for disruption of RAD51 foci formation when FLIP is overexpressed on its own. We have added this new section to the discussion:

“However, it is worth noting that while both proteins likely function in the same pathway, they do not have completely overlapping phenotypes. FIGNL1 depleted cells are not as sensitive to cisplatin as FLIP depleted cells are (Fig 5A). FLIP can interact with RAD51 independently of FIGNL1 (Fig 6C). FLIP overexpression is sufficient to prevent RAD51 foci formation independently of any perturbations to FIGNL1 levels (Fig 7B-C & S7A-B), and lastly, our FLIP-GFP rescue cell line does not fully restore FIGNL1 levels, yet significantly improves viability upon cisplatin treatment and RAD51 dissociation from chromatin (Fig 5B,C and 6I). Nevertheless, in the majority of the phenotypes tested, there were similar outcomes.”

The first report on FIGNL1 function identified it as an HR-promoting protein. It was classified as an anti-recombinase based on in vitro biochemical evidence showing its ability to promote removal of RAD51 from DNA. We believe that proteins that function downstream of RAD51 foci formation are still required for proper recombination either for proper initiation of DNA synthesis after strand invasion and/or for second-end capture during synthesis-dependent strand annealing (SDSA). Thus, without these proteins, completion of HR may not occur, and so we don't see a conflict there.

1. The authors should strongly consider calling RADIF, FLIP. It was already named FLIP in plants and changing the nomenclature is very confusing. The FLIP name makes a lot more sense, “FIGNL interacting protein” whereas RADIF does not. RAD generally relates to radiation sensitivity and this gene does not result in radiation sensitivity, causing even more confusion.

We now provide evidence that RADIF loss leads to mild but significant sensitivity to radiation. However, to the reviewer's point and in order to more closely align with the literature, we have discarded the name RADIF and have updated the text and figures with the name FLIP.

2. The over expression data presented in Figure 3G was unconvincing. It was difficult to observe such small differences in the western blot shown and whether or not that level of potential signaling is biologically significant is unknown. The data should be repeated to show clearer

results, removed, or the language softened significantly. “Taken together, the increased damage signaling observed both when RADIF was lost or overexpressed showed that alterations in RADIF levels was highly correlated with dysregulated DNA damage signaling, suggesting a need for tight regulation of cellular RADIF levels to prevent genomic instability.”

We thank the reviewer for pointing this out. Because of the weak overexpression and the fact that the figure only made a minor point in an unconvincing manner, we have decided to remove the figure. We have added a **new figure 3G** showing increased break formation by neutral comet assay, which we believe strengthens our argument on increased break formation in the absence of FLIP.

3. In Figure 6, RADIF was shown to interact with RAD51. However, it is possible that this interaction is indirect and mediated through FIGNL1. Similarly, the increase in RAD51 foci may also be due to loss of FIGNL1 in the RADIF KO cells. Therefore, the sentence “These results suggest that RADIF regulates dissociation of RAD51 nucleofilaments upon DNA damage and provide a possible mechanism for DNA repair defects seen upon RADIF loss after ICL agent treatment.” should be softened.

We thank the reviewer for pointing this out. We have now investigated this in more detail and added an **updated figure 6A** (using benzonase treatment) as well as two **new figures: 6B and 6C** showing reciprocal IP of RAD51 and FLIP, and FIGNL1-independent interaction with RAD51 respectively. In addition, we have also softened the statement as follows “*These results suggest that FLIP may regulate dissociation of RAD51 from damage foci after DNA damage, providing a possible mechanism for DNA repair defects seen upon FLIP loss after ICL agent treatment.*”

Minor Comments:

1. In Figure 1B-F, the timing of the experiment should be indicated for the different treatments.

We thank the reviewer for pointing out the oversight. We have now added this information.

2. The labeling in Figure 4 is confusing. It is hard to tell what is tagged with what from the figure itself. Were these experiments performed in the presence of DNase? Is Pre-IP, input? Input is a more typical way to refer to the samples.

Yes, Pre-IP is input. We have now changed the labeling to input and added information on when benzonase treatment was performed.

3. Supplementary Figure 5F, the blot is a very poor quality, and it is hard to see if RADIF is being depleted. Was this experiment repeated more than once? A better blot should be included or the data should be removed.

Yes, we agree. Unfortunately, the antibody isn't great, and is particularly unsuited for fractionation experiments. This was our 'best' blot after multiple attempts. We have decided to remove it (New S6F).

4. This sentence is very speculative and doesn't belong in the results. “Taken together, our results demonstrate an HR function for RADIF and suggest that RADIF inhibition might be a viable way to enhance sensitivity of BRCA1 mutant cells to PARP inhibitor treatment.

Yes, we agree with this assessment as well. We have rephrased that section as follows:

“Taken together, our results demonstrate an HR function for FLIP and suggest that FLIP and FIGNL1 together promote viability after exposure to various kinds of DNA damage.”

Reviewer #2 (Remarks to the Author):

This manuscript characterizes the function of C1orf112 in inter-strand cross link (ICL) repair. The authors propose that C1orf112 regulates RAD51 in a manner dependent on its ability to interact with FIGNL1 thereby naming the protein RADIF (RAD51-regulatory Interactor of FIGNL1). RADIF was identified as one of the top scoring genes from 4 previously published datasets screened for cisplatin sensitivity. When RADIF is lost, cells exhibit hallmarks of extensive genome instability characterized by increases in micronuclei formation and chromosomal abnormalities. RADIF localizes to sites of damage but functions in parallel to FA pathway to repair ICLs. Using BioGrid interactome data, the authors investigate interaction of RADIF with FIGNL1 to show that N-terminal regions of both proteins help form a co-stable complex and that FIGNL1 functions like RADIF in repairing ICL lesions. Interestingly, FIGNL1 is downregulated in RADIF-proficient cells after cisplatin exposure. Further characterization reveals that loss of RADIF or FIGNL1 causes increased RAD51 association with chromatin. In cisplatin-treated cells, RAD51 persists post recovery suggesting that RADIF-FIGNL1 complex functions to dissociate RAD51 post nucleofilament assembly to mediate repair. RADIF inactivation causes defects in replication fork elongation and sensitizes BRCA1-deficient SUM149 cells to PARPi.

This is a potentially interesting manuscript describing RADIF function in ICL repair. The authors have pursued several avenues to support their conclusions. However, much of the data included requires further characterization and reproducibility. Specifically, the mechanism by which RAD51 is disassembled is not thoroughly elucidated. Which RAD51-DNA intermediate does RADIF act on? Why is replication fork integrity compromised when RADIF is lost? Unlike RADIF, the genetic dependencies shown for FIGNL1 is limited and requires additional work, using the described truncation mutants, to support the claim that FIGNL1 and RADIF function together to regulate RAD51 in ICL repair.

We thank the reviewer for the positive comments regarding our work. We believe we have now addressed the previous weaknesses of the paper as outlined by the reviewer. Our overall model is that both proteins share majority of their functions, and since prior biochemical characterization of FIGNL1 has shown that FIGNL1 can dissociate RAD51 from DNA, our conclusions regarding RADIF may be warranted. We have also addressed the replication defects with additional data. While our data is not in support of a complete overlap of phenotypes when either protein is lost, we believe the additional evidence examining FIGNL1 in a **largely new Figure 7** and **updated Figure 8** is sufficient to conclude that both proteins function in the same pathway. Specific responses are detailed below.

Major comments:

The authors characterized the interaction between RADIF and FIGNL1 but it's not clear whether FIGNL1 was also identified as a hit from the genetic screens. If FIGNL1/RADIF complex is important for cisplatin repair, why was FIGNL1 not identified? FIGNL1 is not seen in the list. What was the confidence score compared to RADIF? Also, parallel characterization of FIGNL1 is needed to support conclusions derived for RADIF function in the context of results presented in Figure 7. Are RADIF and FIGNL1 functions always co-dependent?

FIGNL1 also scored in the screens. It just didn't meet our stringent filter threshold. There are several genes including several Fanconi members that scored in individual screens but didn't meet the criteria we set.

While the focus of this paper is mainly on the previously uncharacterized RADIF/FLIP, we agree that the experiments in the old Figure 7 ought to be expanded to include FIGNL1. Since prior papers on FIGNL1 have characterized its HR functions, we have limited our analyses to newly described roles in this manuscript like its function in regulating fork progression and the increased sensitivity of BRCA1 mutant cells to FIGNL1 depletion. These have been added as a **new Figure 8C and 8F** (the old figure 7 is now figure 8). We also added a **new figure 7E** showing epistasis between FIGNL1 depletion and FLIP/RADIF loss with regards to RAD51 foci persistence.

Whether the interaction between RADIF and FIGNL1 is sufficient to promote viability to ICL is not thoroughly investigated. Although N228 mutant is unable to rescue viability, it can mean N228 is defective in additional interactions. Authors need to explore RADIF interactome and show changes in other interactions cannot explain the inability of the N228 construct to rescue.

We agree with the reviewer that our experiments are insufficient to make those conclusions. We have removed the initial statement and softened the language in the paper by adding the following caveat:

“While this data suggests that FLIP and FIGNL1 function in the same pathway to mediate cellular viability after exposure to replication stress agents like cisplatin, we cannot rule out that the N228 truncation of FLIP has additional important functions outside of FIGNL1-binding.”

Figure 6A results are not conclusive. RADIF expression seems variable across experiments (compare to Fig 4A where expression is robust). Authors should repeat and test for robust pulldown of RAD51. RAD51 pulldown also seems comparable to EV. A reverse IP is also necessary to confirm interaction. Are these interactions DNA dependent?

We agree that the original set of data were not strong enough. We have now done more extensive analysis of this finding. RADIF overexpression has been very challenging in our hands. Despite the very low expression of RADIF achievable when expressing the protein, we repeatedly observed benzonase-resistant interaction with RAD51. We have **updated figure 6A** with an improved figure that has less background in the vector-only lane and has been done under benzonase treatment conditions, to rule out DNA binding as a mediator of the interaction between RAD51 and RADIF. In addition, we have performed reciprocal experiments showing pulldown of RADIF by RAD51. This was added as a **new figure 6B**. We have also included a new figure showing that RADIF binding to RAD51 was not dependent of FIGNL1. In this experiment, RADIF is overexpressed and surprisingly knockdown of endogenous FIGNL1 did not affect the stability of the overexpressed protein allowing us to demonstrate FIGNL1-independent interaction between RADIF and RAD51. This has been added as a **new figure 6C**. We did attempt to pull down endogenous FLIP protein using RAD51 antibodies but were unsuccessful due to technical challenges (poor antibodies). Taken together we think our new set of experiments improve our conclusion that RADIF binds to RAD51. Future experiments will be aimed at further mechanistic characterization of this interaction including biochemical approaches.

There is some mechanistic ambiguity in terms of how RAD51 is precisely regulated. Why are RAD51 foci higher in untreated RADIF-deficient cells? RADIF-depleted cells exhibit increased DNA damage which could be explained by increased DNA breaks but evidence for this result is not shown. Are the RAD51 foci in untreated conditions observed in all phases or is the increase specific to replication? Data suggesting that dissociation of RAD51 after DNA damage repair is defective is not thoroughly tested. The increase in chromatin bound RAD51 post 72 hours is

reflective of inherently increased presence of RAD51 on chromatin. Experiments showing RAD51 recruitment and dissociation dynamics specifically at break sites is required to support the conclusion of a function in RAD51 dissociation.

We thank the reviewer for the helpful questions /suggestions. We believe we have now addressed most of the reviewer questions. Using EdU labeling and Click chemistry, we have investigated the increased RAD51 foci formation (which we see in about 50% of RADIF null cells without exogenous DNA damage) and shown in a **new Figure 7F,G,H (and supplemental Figure S7C)** that this likely occurs in S phase. We have also included new comet assay data in a **new Figure 3G** showing increased break formation in RADIF null cells in the absence of exogenous DNA damage. We have also more thoroughly tested RAD51 dissociation after DNA damage and shown in an **updated Figure S6C,D** that there are greater differences between WT and RADIF null cells 48 h and 72 h after release from damage compared to the slight change at 0h, consistent with our model. A look into epistasis between both proteins in a **new figure 7E** also supports this conclusion this time examining FIGNL1 co-depletion with RADIF loss. We have also added a **new Figure 7D** showing both FLIP and FIGNL1's ability to promote RAD51 foci persistence at damaged sites in an antagonistic manner to SWSAP1 loss. We think that these experiments strengthen our conclusions and improve the paper.

The authors conclude in Figure 1H/I that RADIF depleted cells undergo increased apoptosis after cisplatin treatment but no evidence is provided to support this claim.

We believe that increased sub-G1 DNA fractions can be inferred to represent increased apoptosis as has been noted by others. PMIDs: 17252584,18603118.

The authors claim that the doses of HU do not damage DNA. 2mM HU is generally considered high-dose that is sufficient to induce breaks within few hours.

We thank the reviewer for this point. We have **removed the statement**.

In figure 3F, increased RPA phosphorylation is only observed in one clone. What is the reason for this discrepancy between the two clones? The reported RADIF clones seems to behave differently.

We believe this was due to a loading error when performing this set of westerns. We have now replaced the figure with an **updated figure (now 3E instead of 3F)**.

Without stable C-terminal truncation mutants for RADIF, it is not possible to conclude only the N-terminus region is involved in interaction with FIGNL1. The background for RADIF in the IP lane is also very high to confidently rule out lack of RADIF interaction with N120 FIGNL1 construct.

We thank the reviewer for this point. We have changed the statement to only indicate that this region is required.

Is the fork slowdown dependent on increased RAD51 association with forks? Are similar effects (fig 6 and 7) observed in FIGNL1 depleted cells? Is RADIF recruited to replication forks to counteract RAD51?

These are great questions, and we were interested in this hypothesis as well. To address this, we tested whether RAD51 inhibition using the inhibitor B02 would rescue the replication defects that we see in the absence of RADIF. Unfortunately, RAD51 has been shown previously to be

required for efficient replication (PMID: 20935632), and in our hands, we consistently observed reduced replication upon even low doses and short pulses of RAD51 inhibition. While we are interested in testing this hypothesis, we think that fully addressing this will require a lot more time and new approaches and is likely beyond the scope of the present publication.

Other comments:

The title of the manuscript mentions 'replication stress' is misleading given the authors observe sensitivity only to a subset of stress inducing agents (MMC, Cisplatin and CPT) and not HU.

We thank the reviewer for this point. We think the reviewer will agree that our new CSA data showing a requirement for RAD51 after HU treatment (**New Figure 1E**) suggests that this is no longer misleading.

Line 120: Which cell line was different? Is it relevant?

RPE1 cells. We have added the information.

Line 122: It seems that 500 was randomly selected as a cut-off.

Yes, we had to choose a cutoff and we choose to go with the top 500 since it was easier. In a library of 18000 genes, 500 corresponds to fewer than the top 3%.

Line 127: Figure 1A: Are the genes listed in a ranking order? What is the combined confidence score for C1orf112 from the 3 screens? How 'high' is very highly? Is it comparable to known FANC genes for example? Numerical details should be included instead of mentioning 'very highly'.

The data has been mined from published screens. We have rephrased the statement to "FLIP was one of the top 150 hits". Since it is hard to infer the likelihood of validation based only on gene ranks in an individual screen, we have gone with overlaps from multiple screens. The screens are from different labs under different conditions, so combined confidence score may not be ideal.

Line 140: What are other forms of DNA damage?

We have modified the language to only say base damage.

Fig S1B, 1D,E,F: Statistical values are not included. RPE1 CSA data seems to include only 1 repeat (error bars are missing?)

The error bars not showing up is a quirk of GraphPad Prism presumably since these are log values. This occurs often. We have now included the statistics.

Line 171: Indicate the low dose of MMS used

We have now added the dose to the main text and figure legend.

Figure S2A: Median is 0 for all conditions. How are the differences quantified to be statistically different?

This is possible using Kruskal-Wallis ANOVA followed by Dunn's multiple comparisons test. We have now added the statistics used. However, addition of a third independent replicate also now results in a different median value.

Figure S2C: Which foci were quantified?

We thank the reviewer for catching this error. FANCI. The figure has been updated.

Figure S2D: The changes in ubiquitinated FANCD2 look very minimal.

Yes, we agree.

Figure 3D: CtIP depletion has higher than 100% of RADIF foci. How is it possible?

The data has been normalized, and 100% in this case is the normalized percentage of RADIF foci in control siRNA treated cells. We realize that this is confusing and have remade the figure using the non-normalized values. We have an added additional replicate as requested by reviewer 3.

Figure 3G: Is the overexpression 2-fold? Quantification to support this statement is not provided. How many times was this experiment repeated? Why did the authors generate 'near-endogenous levels of GFP-tagged RADIF' when the purpose was to test the effects of overexpression? The data seems inconsistent with the rationale.

We thank the reviewer for this point. As this figure was problematic with other reviewers as well, we have now removed the figure and replaced with **new comet assay data**.

Figure 4D: line 272: Statement that expression was 'better' is not supported with quantification. Over how many repeats was it observed? The expected molecular weight bands for the truncations should be indicated. Also include IF images to show localization was not affected for the stable proteins.

We have removed the statement.

Line 284: Details are not provided as to how FIGNL1 participates in HR. Which HR function was disrupted?

We were referring to published papers on FIGNL1. We have attempted to make this clearer in the text.

Line 308: Survival is only reflected in cells treated with cisplatin and MMC suggesting it not generalizable to all kinds of replication stress.

We think this has now been addressed above. We see sensitivity to HU using clonogenic survival assays.

Figure 5B: Corresponding western blot is not included. Does the RADIF N228 mutant localize to damage sites?

We have now included the western blot as a **new Figure 5C**.

Figure 5C-F: Did the authors test whether FIGNL1 levels are restored when mutants of RADIF are re-expressed?

We tested this, and to our surprise we only saw a minimal restoration of FIGNL1 levels. The reason for this is not yet clear.

Figure 7D: Changes in DR-GFP signal are very minimal.

We thank the reviewer for pointing this out. It was around a 30% reduction which is significant but is definitely not as strong as you'd expect if reducing core HR factors like BRCA2. We have removed the weaker siRNA.

In figure 3C and 3D, the authors show that FANCD2 loss reduces RADIF foci formation but CtIP loss does not. Is foci formation disrupted in the absence of BRCA2?

We unfortunately did not have the time to test this at this time because our focus was to determine whether RADIF recruitment was dependent on the Fanconi pathway.

Reviewer #3 (Remarks to the Author):

This manuscript describes the functional analysis of a poorly characterized human protein, termed RADIF, which was identified in previous genome-wide screens for factors that confer resistance to the DNA cross-linking agent cisplatin. The authors demonstrate that loss of RADIF leads to accumulation of DNA damage, increased chromosomal instability and altered cell cycle progression upon exposure of cells to cisplatin. In addition, cells lacking RADIF displayed inefficient DSB repair by homologous recombination (HR) and reduced replication fork velocity as compared to normal cells. Further analysis revealed that RADIF accumulates at nuclear foci in response to cisplatin treatment and forms a complex with FIGNL1, which functions as an anti-recombinase by dismantling RAD51 filaments. Moreover, RADIF was found to act in a common pathway with FIGNL1 to confer resistance to cisplatin, and its deficiency led to RAD51 accumulation on chromatin both in the absence and presence of exogenously induced DNA damage. Based on these results, the authors propose that RADIF is essential for limiting RAD51 levels on undamaged chromatin and for dissociation of RAD51 nucleofilaments to properly complete HR. Overall, the data presented are interesting, but most of the key experiments were performed only twice, and in some cases even only once, which casts doubt on the entire study. Moreover, further experiments are needed to gain mechanistic insights into the role of RADIF in RAD51 regulation.

We thank the reviewer for the positive reviews of our work. We have now ensured that the key experiments were performed at least three times, thereby strengthening our conclusions. We have also added some new experiments highlighted below to further address RADIF's role in RAD51 regulation.

Specific points:

1. Experiments must be repeated at least three times (clonogenic/viability assays, measurements of chromosomal aberrations/micronuclei, quantifications of nuclear foci, DR-GFP assays).

We have now **performed all of these and other related key experiments at least three times**. We have **added information regarding the statistics used** to reflect this. We have also **updated all our graphs to show individual points** where applicable.

2. Based on the results of multicolor competition assays presented in Fig. 1B, the authors conclude that RADIF-depleted cells are hypersensitive to cisplatin, but not to hydroxyurea (HU) or ionizing radiation (IR). In the same experiment, they found that HU did not compromise the viability of FANCD2-depleted cells, which is not in agreement with clonogenic survival data in the literature (e.g. 10.1093/nar/gkx847). Therefore, sensitivity of RADIF-deficient cells to HU and IR should be also tested by clonogenic survival assay.

We thank the reviewer for the helpful suggestion. We also wondered about that, and we have now performed CSAs showing sensitivity to HU and surprisingly modest but significant sensitivity to IR as well. These have been included as **new Figures 1E and 1F**. The old figure 1E and 1F have been moved to the supplement.

3. The authors state in the manuscript that RADIF is critical for ICL repair (e.g. page 4, line 104 or page 6, line 189). What is the evidence for this claim? It is possible that ICL agents induce cell death not only through the generation of DNA crosslinks (e.g. cisplatin or MMC increase the cellular levels of ROS). RADIF could be also involved in ICL traverse. In fact, the authors found that RADIF promotes cell survival after cisplatin treatment independently of the FA pathway that

is implicated in ICL repair (Fig. 2D).

We thank the reviewer for this point. We have softened the statement from critical to important.

4. The statement “RADIF and promotes repair following DNA damage” (page 6, line 217) is not supported by the data presented in Fig. 3.

We have **updated Figure 3** with new data showing persistent break formation (**new comet assay data Figure 3G**), damage signaling (**updated Figure 3E**) in the absence of RADIF. This in conjunction with our **updated figure 1 (new figures 1E and 1F)** we believe is consistent with this statement.

5. The western blot data in Fig. 3E-G showing the effect of RADIF depletion/overexpression on the cellular levels of DNA damage markers upon cisplatin treatment are not very convincing. For example, phosho-RPA is increased in delta8 clone but not in delta10 clone (compare lanes 4-6 in Fig. 3F). In any case, only minor differences are observed. Could the authors quantify IF images (QIBC analysis)?

We agree with the reviewer that this was not of the highest quality. We believe that this was due to loading issues in this set of experiments. We have now updated the figure with a **new Figure (now 3E instead of 3F)**. Because of the weak overexpression and the fact that the old figure 3G only made a minor point in an unconvincing manner, we have decided to remove the figure. We have added a **new figure 3G** showing increased break formation by comet assay, which we believes strengthens our argument on increased break formation in the absence of FLIP.

6. Is the accumulation of RADIF foci after cisplatin treatment cell cycle dependent? Co-staining with PCNA antibody or EdU pulse-labeling could be performed to address this question.

We tested this using EdU labeling, and it was not.

7. Fig. 4: Interaction between RAD51 and FIGNL1 should be confirmed by co-immunoprecipitation assay of endogenous proteins. To test whether the interaction is direct, a pull-down assay with purified recombinant proteins could be performed. For Fig. 4C, reciprocal co-IP should be performed to confirm the negative effect of cisplatin treatment on FIGNL1-RADIF complex formation. For Fig. 4D, the blot of IPed RADIF variants should be added. For Fig. 4E, the presence of RADIF band in lane 3 (IP) is not clearly seen.

This has previously been done in the literature PMID 23754376. We were able to repeat the experiment, but our data has high background that isn't publication quality.

Re 4C, we have now done this, and we surprisingly saw a different result doing the experiment this way. This has been included as a **new Figure S4A**. The reason for this discrepancy is not clear and we have softened our language to reflect this.

Unfortunately, antibody cross-reactivity meant we repeatedly could not get a publication-quality blot showing the IP'ed RADIF variants. We tried many ways to clean up the blots, but they failed. However, the IPs were successful.

8. Fig. 5B: The lack of complementation of the cell survival defect of RADIF knockout cells with RADIF deltaN228 mutant does not necessarily mean that the defect is caused by loss of RADIF-FIGNL1 interaction. The deleted N-terminal region (large deletion, 227 aa) could also have another function(s) that is required for cell viability.

Yes, this point was also made by another reviewer, and we agree. We have now deleted our initial statement and softened our language by including this caveat:

“While this data suggests that FLIP and FIGNL1 function in the same pathway to mediate cellular viability after exposure to replication stress agents like cisplatin, we cannot rule out that the N228 truncation of FLIP has additional important functions outside of FIGNL1-binding.”

9. Based on the data shown in Fig. 5 C and D, the authors conclude that FIGNL1 is targeted for proteasomal degradation upon cisplatin treatment. However, the levels of ectopically expressed FIGNL1 are not affected by cisplatin treatment (Fig. 4C). Could the authors explain this discrepancy? If cisplatin treatment causes FIGNL1 degradation, how might RADIF and FIGNL1 function as a complex to prevent cisplatin-induced cell death?

We are grateful to the reviewer for this point, and we agree with the reviewer. The reduction we observed was incomplete. However, our failure to observe such changes in 293T cells coupled with our reverse coIP (Figure S5A) not showing reduced amounts of FIGNL1 nor reduced interaction with FLIP, further added to the conundrum. Because we don't have an answer as of yet, we have decided to remove these two figures since we have more work to do beyond the scope of this present publication to fully understand regulation of FIGNL1 expression upon DNA damage across multiple cell lines. We believe that removal of the figures does not affect the paper's conclusions and makes our model more consistent, and we have updated the discussion to reflect this.

10. Fig. 6A: co-IP data showing RADIF-RAD51 interaction are not very convincing (RAD51 pull-down with EV is also seen). Moreover, co-IP is done again only with ectopically expressed proteins.

This was also mentioned by reviewer 2. We agree that the original set of data were not strong enough. We have now done more extensive investigation. Unlike FIGNL1, there is generally very little over-expression of RADIF achievable in these transfections, so we don't believe that the interaction is an artifact of ectopic expression. That said, we have now **updated figure 6A** with an improved figure that has less background in the vector-only lane and, importantly, has been done under benzonase treatment conditions, to rule out DNA binding as a mediator of the interaction between RAD51 and RADIF. Beyond this, we have performed reciprocal experiments showing pulldown of RADIF by RAD51. This was added as a **new figure 6B**. We have also included a new figure showing that RADIF binding to RAD51 was not dependent of FIGNL1. In this experiment, knockdown of endogenous FIGNL1 did not affect the interaction between RADIF and RAD51. This has been added as a **new figure 6C**. Taken together we think our new set of experiments improve our conclusion that RADIF binds to RAD51. We did attempt to pull down endogenous RADIF protein using RAD51 antibodies but were unsuccessful due to technical challenges (very high background due to poor antibodies). Future experiments will be aimed at further mechanistic characterization of this interaction including biochemical approaches.

11. The authors should examine whether RAD51 foci colocalize with RADIF foci following cisplatin treatment to further support the conclusion that RADIF physically interacts with RAD51 and regulates its levels on chromatin.

We thank the reviewer for this suggestion. We attempted to do this but were unsuccessful due to technical reasons. Our validated RAD51 antibody that works well is of rabbit origin and so is our validated GFP antibodies. We obtained and tested a few mouse GFP antibodies but were unable to detect GFP above background levels.

12. The authors conclude that RADIF acts in conjunction with FIGNL1 anti-recombinase to regulate RAD51 filament assembly. Cells lacking the RAD51 paralog SWSAP1 have been shown to exhibit defective RAD51 focus formation, which is suppressed by FIGNL1 depletion (DOI: 10.1038/s41467-019-09190-1). Does RADIF depletion also suppresses this phenotype of SWSAP1-depleted cells? In addition, the effect of FIGNL1 depletion on the level of RAD51 foci in RADIF KO cells should be evaluated to determine whether these proteins act in a common pathway. The results of these experiments can provide strong support for the above hypothesis.

We thank the reviewer for these very helpful suggestions. We agree that given our model both of these questions should be tested. We have now done so. Consistent with our model, RADIF loss phenocopies loss of FIGNL1 with respect to SWSAP1. This has now been included as a **new Figure 7D** (SWSAP1 experiment). Likewise in a **new Figure 7E** (FIGNL1 RADIF KO co-depletion experiment) we show that FIGNL1 depletion does not further affect RAD51 levels in RADIF KO cells.

13. The authors suggest that the observed reduction in replication fork velocity in RADIF-deficient cells might be a consequence of increased levels of RAD51 on chromatin, causing fork reversal. This hypothesis should be tested by experimentally, for example by depleting ZRANB3 translocase, which mediates fork reversal. It will be also interesting to test whether replication fork progression is compromised by FIGNL1 depletion.

These are great suggestions, and we were interested in this hypothesis as well. Our thinking was that increased fork reversal will lead to slowing down of replication, thus reducing fork reversal should lead to rescue of replication progression defects. To address this, we depleted ZRANB3 alone or in combination with RADIF and repeated the experiments. What we found was that ZRANB3 knockdown did not rescue the progression defect and surprisingly led to further reduction in fork velocity. The reason for this is not yet clear. We also inhibited RAD51, but this caused reduced red fiber lengths compared to control cells, complicating our ability to interpret whether RAD51 inhibition can rescue RADIF loss-induced fork shortening. We think there needs to be a lot more work to fully address what is going on that is beyond the scope of this present publication and so we have not included this data in the paper. However, we have included the ZRANB3 data below for the reviewer. We will perform more work in future to address this.

Re FIGNL1, we have tested FIGNL1 knockdown effects and observed similar phenotypes as with RADIF loss. This data has been included as a **new Figure 8C**. (The old figure 7 is now Figure 8).

REVIEWERS' COMMENTS

Reviewer #1 (Remarks to the Author):

The authors have now addressed my concerns. I appreciate their thoughtful revision.

Reviewer #2 (Remarks to the Author):

The authors have fully addressed comments on the previous submission, which have improved the quality of the manuscript. The clarifications and additional experimental results provided by the authors have addressed my concerns.

Reviewer #3 (Remarks to the Author):

The authors adequately addressed the majority of my specific comments but they did not consider my minor points:

1. Page 3, lines 78-82: Please insert citations.
2. Page 3, lines 91-93: Please insert citations.
3. Fig. 2D: the y-axis should be labeled "Cell Survival (%)". The same applies to Fig. 5A and 5B.
4. Page 27, line 982: "chromosomal instability" would be more appropriate here.
5. Page 10, line 406: "... promotes replication fork progression" is more appropriate here.
6. Page 11, lines 432-434: The authors state: "Consistent with a requirement for RADIF for proper HR, MCA assays showed that RADIF knockdown led to reduced viability after camptothecin (CPT) treatment ..." It should be noted that the sensitivity of cells to CPT does not necessarily reflect a defect in HR". CPT also induces formation of R-loops which can lead to replication fork collapse.
7. Micronucleus assay should be described in Methods.
8. Fig. 7A and B: It should be indicated in the schematic when CD437 was added to cells. CD437 concentration used should be described in figure legends.

POINT-BY-POINT RESPONSE REVIEWERS' COMMENTS

Reviewer #1 (Remarks to the Author):

The authors have now addressed my concerns. I appreciate their thoughtful revision.

We thank the reviewer for their helpful suggestions for improving our manuscript.

Reviewer #2 (Remarks to the Author):

The authors have fully addressed comments on the previous submission, which have improved the quality of the manuscript. The clarifications and additional experimental results provided by the authors have addressed my concerns.

We thank the reviewer for their helpful suggestions for improving our manuscript.

Reviewer #3 (Remarks to the Author):

The authors adequately addressed the majority of my specific comments but they did not consider my minor points:

We thank the reviewer for their helpful suggestions for improving our manuscript. We apologize for the copying error that led to us failing to initially address the reviewer's minor points. These have all now been addressed.

1. Page 3, lines 78-82: Please insert citations.

Done.

2. Page 3, lines 91-93: Please insert citations.

Done.

3. Fig. 2D: the y-axis should be labeled "Cell Survival (%)". The same applies to Fig. 5A and 5B.

Done.

4. Page 27, line 982: "chromosomal instability" would be more appropriate here.

Rephrased.

5. Page 10, line 406: "... promotes replication fork progression" is more appropriate here.

Rephrased.

6. Page 11, lines 432-434: The authors state: “Consistent with a requirement for RADIF for proper HR, MCA assays showed that RADIF knockdown led to reduced viability after camptothecin (CPT) treatment ...” It should be noted that the sensitivity of cells to CPT does not necessarily reflect a defect in HR”. CPT also induces formation of R-loops which can lead to replication fork collapse.

We have added the following caveat: “*However, since CPT also induces formation of R-loops that can lead to fork collapse (PMID: 25435140), this phenotype may not necessarily reflect HR defects.*”

7. Micronucleus assay should be described in Methods.

Done.

8. Fig. 7A and B: It should be indicated in the schematic when CD437 was added to cells. CD437 concentration used should be described in figure legends.

Done.